# SPUS: A Lightweight and Parameter-Efficient Foundation Model for PDEs

## Abstract

We introduce Small PDE U-Net Solver (SPUS), a compact and efficient foundation model (FM) designed as a unified neural operator for solving a wide range of partial differential equations (PDEs). Unlike existing state-of-the-art PDE FMs—primarily based on large complex transformer architectures with high computational and parameter overhead—SPUS leverages a lightweight residual U-Net-based architecture that has been largely underexplored as a foundation model architecture in this domain. To enable effective learning in this minimalist framework, we utilize a simple yet powerful auto-regressive pretraining strategy which closely replicates the behavior of numerical solvers to learn the underlying physics. SPUS is pretrained on a diverse set of fluid dynamics PDEs and evaluated across 7 challenging unseen downstream PDEs spanning various physical systems. Experimental results demonstrate that SPUS using residual U-Net based architecture achieves state-of-the-art generalization on these downstream tasks while requiring significantly fewer parameters and minimal fine-tuning data, highlighting its potential as a highly parameter-efficient FM for solving diverse PDE systems.

## 1 Introduction

Partial differential equations (PDEs) are fundamental mathematical tools for modeling a wide range of complex spatio-temporal phenomena in science and engineering, including fluid dynamics, electromagnetism, materials science, and climate systems (Neumann et al., 2012; Schaa et al., 2016; Müller & Scheichl, 2014). Traditional numerical solvers—such as finite difference and finite element methods—are widely used for PDE simulation but often come with high computational costs, especially when repeated simulations are required for varying coefficients or boundary conditions (Herde et al., 2024). To address these limitations, deep learning–based approaches like the Fourier neural operator (Li et al., 2021), convolutional neural operator (Raonic et al., 2023), and DeepONet (Lu et al., 2021) have been proposed. While these models have shown promising performance, they are typically designed for specific PDE families and require retraining when applied to new classes of governing equations, leading to significant computational overhead.

Some simulation data is more computationally expensive to produce from numerical solvers than others; and so multiphysics PDE FMs take advantage of pretraining on large benchmark PDE data to finetune on limited PDE data from more expensive simulations. PDE FMs—including MPP (McCabe et al., 2024), POSEIDON (Herde et al., 2024), PROSE FD (Liu et al., 2024), DISCO (Morel et al., 2025) and DPOT (Hao et al., 2024)—have emerged as a promising paradigm. These models aim to learn unified representations by incorporating multiple physical systems into a single framework, demonstrating the ability to generalize to unseen PDE families using limited data. However, current state-of-the-art FM approaches predominantly utilize transformer-based architectures with high parameter counts in the hundreds of millions, resulting in increased computational and data demands (McCabe et al., 2024; Herde et al., 2024; Hao et al., 2024). To overcome these limitations, in this work, we propose a efficient yet effective Small PDE U-Net Solver (SPUS), with an order of magnitude fewer parameters, for PDE foundation modeling. To the best of our knowledge, this is the first work to explore a residual U-Net architecture as an FM pretrained on a large and diverse PDE dataset, beyond single-family PDE prediction. U-Net has been shown to significantly outperform neural operators such as FNO in solving PDEs (Gupta & Brandstetter, 2023). However, recent transformer-based FM approaches primarily compare against U-Net on a single family

of PDEs (McCabe et al., 2024; Hao et al., 2024; Shen et al., 2024), overlooking its potential as a foundation model architecture—particularly given the availability of large-scale PDE datasets from diverse systems.

PDE FMs are formulated in various ways with different assumptions on the form of input data and output predictions. Other FMs rely on multiple previous timesteps as input and predict output trajectories, or include temporal information in the input (McCabe et al., 2024; Herde et al., 2024; Hao et al., 2024). SPUS is trained autoregressively as an operator predicting a single timestep; which closely replicates the behavior of numerical solvers; potentially learning the underlying physics (Lippe et al., 2023). We demonstrate that the autoregressive training produces a foundation model that generalizes to time-independent PDEs via finetuning.

This work addresses the following key questions regarding FMs for PDEs:

(a) Rather than designing a new and complex architecture, can we utilize a simple, existing one—such as residual U-Net—as an FM for PDEs?

(b) Can a lightweight, low-parameter FM achieve state-of-the-art generalization on unseen PDEs?

(c) Can pretraining on a set of simpler PDEs but exhibiting diverse physical behaviors (e.g., shocks, shear, vorticity) enable effective transfer to downstream tasks governed by complex PDEs and dominant dynamics, such as vortex evolution from piecewise-constant or shear-layer initial conditions?

(d) Can an FM be pretrained to emulate the behavior of numerical solvers by autoregressively predicting the next time step from the current one, thereby potentially learning the underlying physics?

Finally, we demonstrate that SPUS, built on a simple residual U-Net and pretrained to emulate the behavior of numerical solvers, achieves state-of-the-art generalization with a lightweight design, transfers knowledge effectively from simpler PDEs to more complex ones, and thereby establishes a path toward efficient, generalizable PDE foundation models.

## 2 Preliminaries

PDEs model a wide range of physical phenomena and include equations such as Navier-Stokes, compressible Euler, the Wave equation and others. The general form of a time-dependent PDE is:

$$\delta_t u(p,t) + L(u, \nabla_p u, \nabla_p^2 u, \ldots) = 0, \qquad \forall p \in D \subset \mathbb{R}^d, \ t \in (0, T),$$

$$\mathcal{B}(u) = 0, \qquad \forall (p,t) \in \delta D \times (0, T), \qquad (1)$$

$$u(p,0) = u_0(p), \quad p \in D.$$

for given boundary conditions $\mathcal{B}$ and initial conditions $u_0$. Note that we use atypical PDE notation with $p$ for the position variable (reserving $x$ for the input as is typical in machine learning). Many PDE datasets are discretized in space and time. We denote the discretized spatial state at each timestep as $u_t = \{(p^j, u_t^j) : p^j \in \mathcal{P}\}, t \in [0, 1, \ldots, n]$ where $\mathcal{P}$ is the discretized spatial mesh and $n$ is the number of discretized timesteps. Initial conditions are given by $u_{t=0}$ and each $u_t \in \mathbb{R}^d$ where $d$ is the dimensionality of system variables.

## 3 Relevant Work

The closest PDE FMs to ours fall into three distinct formulations.

(a) PDE FMs which take $\{u_{t=[0,m]}\}$ of a PDE trajectory as input and autoregressively predict $\{u_{t=[m+1,n]}\}$; where $m = 15$ for MPP (McCabe et al., 2024), and $m = 9$ for DPOT (Hao et al., 2024). MPP projects normalized field variables from diverse physical systems into a unified latent space and utilizes an axial attention vision transformer-based architecture to perform autoregressive prediction over multiple systems. On the other hand, DPOT injects small-scale noise to $\{u_{t=[0,m]}\}$ and utilize a Fourier attention based transformer architecture to perform autoregressive prediction over multiple systems.

(b) PROSE FD (Liu et al., 2024) which takes $\{u_{t=[0,m]}\}$ of a PDE trajectory as input and simultaneously predict $\{u_{t=[m+1,n]}\}$ as a trajectory where $m = 9$ and $n = 19$. PROSE FD introduces a multimodal transformer framework which takes $\{u_{t=[0,m]}\}$ of a PDE trajectory and mathematical description of the physical behavior as input and performs simultaneous prediction for multi-physics systems.

(c) POSEIDON (Herde et al., 2024) which takes $(u_{t=0}, \Delta t)$ as input and predicts $\{u_{t=\Delta t}\} \forall \Delta t \in [1, T]$ where $T = 14$. POSEIDON proposes a multiscale operator transformer architecture enhanced with time-conditioned normalization to perform prediction on multiple physical systems. Similar to SPUS, POSEIDON uses only a single time step (rather than a trajectory) as input; however, instead of performing autoregressive rollout, it predicts arbitrary future time steps directly. For a dataset with $n$ time steps, POSEIDON trains on $O(n^2)$ input-output pairs, whereas our approach is more sample-efficient, requiring only $O(n)$ sequential pairs.

## 4 METHODS

SPUS is a lightweight, low-parameter residual U-Net architecture designed for modeling PDE dynamics. To enable effective learning within this compact model, we utilize an auto-regressive pretraining scheme. This method facilitates the efficient modeling of temporal dynamics of PDEs with reasonable accuracy and low computational overhead.

**Problem statement**   Given an initial state $u_{t=0}$ of a trajectory governed by a specific PDE, where $u_t \in \mathbb{R}^d$ represents the system state at time step $t$ with $d$ variables, our objective is to predict the future states $u_{t=1}, u_{t=2}, \ldots, u_{t=n}$.

**Auto-regressive pretraining and finetuning**   We formulate the problem as a *first-order Markov process* (Pillai, 2002), in which the evolution of the system depends only on the immediately preceding state. That is, the prediction of $u_{t+1}$ is conditioned solely on $u_t$, satisfying the Markov property:

$$P(u_{t+1} \mid u_t, u_{t-1}, u_{t-2}, \ldots, u_0) = P(u_{t+1} \mid u_t). \tag{2}$$

This formulation allows the system dynamics to be modeled using an autoregressive framework consistent with the Markov assumption.

The proposed auto-regressive training methodology for the U-Net-based FM is illustrated in Figure 1. During pretraining, the proposed FM takes a randomly sampled ground-truth state $u_t$ from a PDE trajectory in the pretraining dataset and predicts the next state $u'_{t+1}$. More specifically, during pretraining, only ground-truth states are used as inputs; predicted states are not used to generate future predictions. During finetuning, the pretrained model is adapted to a specific downstream PDE using the same input-output structure as in pretraining: the model receives $u_t$ and predicts $u'_{t+1}$. At inference time, however, we provide the model with the initial state $u_{t=0}$ and auto-regressively generate predictions $u'_{t=1}, u'_{t=2}, \ldots, u'_{t=n}$, where each prediction $u'_{t+1}$ is based on the previously predicted state $u'_t$ as shown in Figure 1.

### 4.1 MODEL ARCHITECTURE

Figure 2 shows the residual U-Net architecture (Lan & Zhang, 2020; Ronneberger et al., 2015) with 36 million parameters we have utilized for designing the FM for PDEs. The U-Net model takes any current state $u_t$ of shape $d \times 128 \times 128$, where $d$ is the number of system variables and applies a $3 \times 3$ convolutional layer to project it into a 32-dimensional feature space. The residual encoder path comprises four hierarchical levels, each of which processes features through two residual blocks. Each residual block includes two $3 \times 3$ convolutional layers, with batch normalization (Bjorck et al., 2018) and GELU activation (Hendrycks & Gimpel, 2016) applied after each convolution, and incorporates a skip connection to preserve feature integrity and support gradient flow. Strided convolution is applied for downsampling at each label of encoder except the last. The encoder processes features through 32-channel blocks at Level 0, increases to 64 channels at Levels 1 and 2, and reaches 128 channels at Level 3. The residual bottleneck consists of two residual blocks that operate on 128-channel feature maps, effectively capturing high-level representations. The residual decoder mirrors

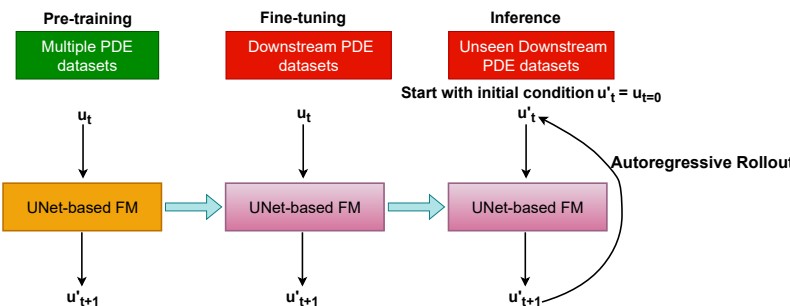

Figure 1: Proposed auto-regressive training methodology for the U-Net-based FM. During both pre-training and finetuning, the FM randomly samples a ground truth state $u_t$, where $u_t \in \mathbb{R}^d$ represents the system variables at time step $t$, and learns to predict the next state $u'_{t+1}$. During inference, the full trajectory is predicted autoregressively from the initial condition $u_{t=0}$. The FM takes $u'_t = u_{t=0}$ as input and recursively predicts subsequent states based on its own previous outputs for $t = 1, \ldots, n$, where $n$ is the length of the trajectory under consideration.

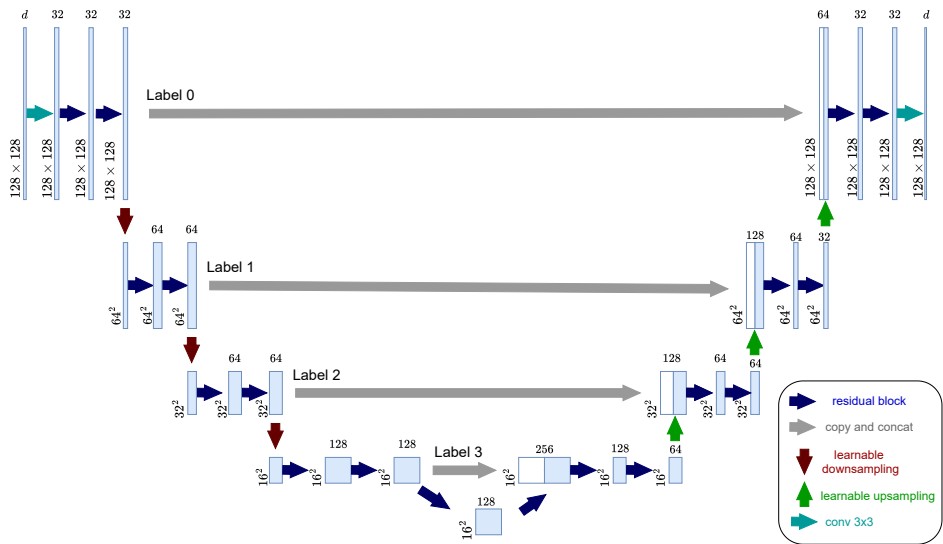

Figure 2: Illustration of the residual U-Net based FM architecture for PDEs with 36M parameters. The network takes an input of shape $d \times 128 \times 128$, representing the current time step of a PDE trajectory, and predicts the next time step of the same shape. It employs an encoder–decoder structure with residual blocks, skip connections, and progressive downsampling and upsampling to preserve spatial and contextual information.

the encoder with three upsampling stages implemented with transposed convolution layers. At each stage, the upsampled features are concatenated with the corresponding encoder feature maps via skip connections, facilitating the recovery of spatial details. After concatenation, the features are passed through two residual blocks, each followed by batch normalization and GELU activation. The decoder progressively reduces the feature dimensionality from 128 to 64, and subsequently from 64 to 32 across its stages. Finally, a $3 \times 3$ convolution maps the decoder output back to the number of system variable $d$.

During pretraining, the model was trained for 200 epochs using the Adam optimizer (Kingma & Ba, 2014) with a linear learning rate schedule starting from $10^{-4}$, and a batch size of 10. The learning rate decreased linearly over the course of training. The model achieving the best performance on the evaluation set of the pretraining dataset was saved for downstream use.

## 4.2 THEORETICAL FOUNDATIONS OF RESIDUAL U-NET EFFICIENCY AND GENERALIZATION IN PDE LEARNING

Design choices are driven by theoretical motivation: a) convolutional layers for spatial bias, b) residual blocks for numerical stability, c) U-Net architecture for encoder-bottleneck-decoder compact representation, and d) multi-PDE pre-training for generalization.

**Convolutional Layers** The step-to-step evolution of nonlinear PDEs on uniform grids exhibits a locally translation-equivariant structure: each grid cell interacts primarily with its spatial neighbors through couplings that decay smoothly with distance (LeVeque, 2007). This locality aligns naturally with the inductive bias of the residual U-Net architecture, whose convolutional encoder captures spatially local dependencies and whose decoder reconstructs fine-scale features through upsampling and skip connections (Ronneberger et al., 2015; Ruthotto & Haber, 2020). By progressively reducing and restoring spatial resolution, the U-Net builds a **multiscale hierarchy** that aggregates coarse and fine spatial information, enabling efficient representation of both local interactions and long-range dependencies. This hierarchical design allows the network to approximate global spatial couplings without the quadratic computational overhead of transformer-based attention mechanisms.

**Residual Blocks** Residual connections enhance both stability and learning efficiency by predicting incremental field updates rather than full mappings, analogous to time-stepping schemes in numerical PDE solvers. These design principles collectively make the residual U-Net a parameter-efficient and numerically stable architecture for learning complex, nonlinear PDE operators across diverse physical regimes.

The following section formalizes a theoretical error decomposition, illustrating how the residual encoder–bottleneck–decoder hierarchy reduces approximation, projection, and statistical errors, and how multi-PDE pretraining further enhances cross-family generalization.

### 4.2.1 THEORY

**Encode-Bottleneck-Decoder Architecture** Let $\Psi : X \to Y$ denote the one-step solution operator of a time-dependent PDE. Given a field state $u_t \in \mathbb{R}^d$ at time $t$, representing the spatial distributions of system variables (e.g., density, velocity, or pressure), the operator $\Psi$ produces the evolved field $u_{t+1} = \Psi(u_t) \in Y$ at the next timestep. For notational convenience, we denote $x = u_t \in X$ as the input field and $y = \Psi(x) = u_{t+1} \in Y$ as the corresponding evolved field. In the residual U-Net framework, this mapping is approximated by a multiscale neural surrogate $F_\theta : X \to Y$, parameterized by weights $\theta$, where each stage of the network (encoder, bottleneck, and decoder) performs local residual updates that emulate incremental field evolution consistent with the underlying PDE dynamics across spatial scales. Let $\mu$ denote the probability measure associated with the sampling of input fields $x$ at time $t$ within the function space $X$. Each sample $x \sim \mu$ therefore represents a physical state of the PDE system at the previous timestep, and the network is trained to predict the corresponding evolved field $y = \Psi(x)$ at time $t + 1$, yielding the expected mean-squared prediction error

$$\mathcal{E}(\theta) = \mathbb{E}_{x \sim \mu} \left[ \| \Psi(x) - F_\theta(x) \|_2^2 \right], \tag{3}$$

which quantifies the average discrepancy between the true PDE evolution and the residual U-Net prediction across all field states encountered in the data. Following the generalization framework of Bhattacharya et al. (2021), this error can be decomposed into three fundamental contributions—neural approximation, model reduction, and finite-sample statistics:

$$\mathcal{E}(\theta) \lesssim \underbrace{\varepsilon_{\text{approx}}^2}_{\text{neural approximation}} + \underbrace{R^\mu(V_x) + R^{\Psi \# \mu}(V_y)}_{\text{model reduction / projection}} + \underbrace{\mathcal{O}\left(\frac{1}{\sqrt{N}}\right)}_{\text{finite-sample term}}. \tag{4}$$

Here, neural approximation denotes the best achievable error of the residual U-Net within the learned subspaces; $V_x \subset X$ and $V_y \subset Y$ denote the encoder and decoder subspaces, respectively. The projection terms quantify the residual energy of the data distributions outside these learned subspaces. Specifically, $R^\mu(V_x) = \mathbb{E}_{x \sim \mu}[\|x - \Pi_{V_x}(x)\|_2^2]$ measures the expected reconstruction error of input fields $x \in X$, where $\Pi_{V_x}$ denotes the projection or learned feature mapping induced by the encoder onto the subspace $V_x$. Likewise, $R^{\Psi \# \mu}(V_y) = \mathbb{E}_{y \sim \Psi_\# \mu}[\|y - \Pi_{V_y}(y)\|_2^2]$ measures the corresponding residual energy of output fields $y = \Psi(x) \in Y$, where $\Psi_\# \mu$ is the output distribution induced

by the PDE operator $\Psi$, that is, the distribution of evolved fields obtained by mapping input samples $x \sim \mu$ through $\Psi$. Here, $\Pi_{V_y}$ denotes the decoder-side projection onto the subspace $V_y$, and $N$ represents the number of training samples. In a residual U-Net, the encoder, bottleneck, and decoder all contain residual blocks of the form

$$h^{(l+1)} = h^{(l)} + R_{\theta_l}^{(h)}(h^{(l)}), \tag{5}$$

where $h^{(l)}$ denotes the feature tensor at depth $l$ *within the current stage*, and each residual operator $R_{\theta_l}^{(h)}$ is a local convolutional transformation acting on the corresponding feature space. The **residual encoder** $E$ progressively constructs a hierarchical latent subspace $V_x$ that captures coarse-to-fine spatial and dynamical modes. By composing small residual updates, the encoder acts as a data-driven reduced-basis generator, minimizing the input projection error $R^\mu(V_x)$ in equation 4. At the network's core, the **residual bottleneck** $B$ captures cross-scale and nonlocal interactions in the latent feature space, providing an efficient, low-dimensional approximation of global coupling that further reduces the intrinsic neural approximation error $\varepsilon_{\text{approx}}$. The **residual decoder** $D$ then reconstructs the output subspace $V_y$ through a sequence of incremental updates, which refine hierarchical features and approximate stable step-to-step updates of the underlying PDE evolution. The complete network composition is thus

$$F_\theta = D \circ B \circ E = (\text{Id} + R_\theta^{(D)}) \circ (\text{Id} + R_\theta^{(B)}) \circ (\text{Id} + R_\theta^{(E)}), \tag{6}$$

where $\text{Id}$ denotes the identity mapping on the corresponding feature space of each component (encoder, bottleneck, or decoder). This composition can be interpreted as a **multiscale residual integrator** in feature space, in which each component performs a small, well-conditioned update aligned with the underlying PDE dynamics. Combining Equations 5–6 with the error decomposition in Equation 4, we find that the residual encoder–bottleneck–decoder hierarchy reduces all three error sources simultaneously: (a) the multiscale residual encoder–decoder minimizes the projection errors $R^\mu(V_x)$ and $R^{\Psi\#\mu}(V_y)$; (b) the bottleneck and residual formulation lower the intrinsic approximation error $\varepsilon_{\text{approx}}$ by capturing local-to-global nonlinear interactions; and (c) convolutional weight sharing limits parameter growth, thereby reducing the statistical term $\mathcal{O}(N^{-1/2})$. In contrast to transformer-based operator learners that rely on global self-attention—incurring quadratic computational and memory complexity with respect to input size and requiring large parameter counts to infer spatial locality from data—the residual U-Net embeds these spatial priors directly through its convolutional and multiscale hierarchical design. This inductive bias enables the residual U-Net to achieve comparable or superior accuracy with substantially fewer parameters and improved numerical stability in PDE operator learning.

**Multi-PDE Pretraining**   When the residual U-Net is pretrained across multiple PDE families $\{\Psi_m\}_{m=1}^M$ defined on the same grid class, the model learns a **shared multiscale encoder–bottleneck–decoder representation** that captures structural invariants common to these operators, including spatial locality, smooth spectral decay, and hierarchical coupling across scales. Assuming that these structural invariants exist in the downstream finetuning PDEs, then model error will be reduced compared to training from scratch (Bhattacharya et al., 2021; Ben-David & Borbely, 2008). Let $\mu_m$ denote the probability measure associated with sampling the variable fields at time $t$ from trajectories of the $m$-th PDE family within the function space $X$, and let $\Psi_{m\#}\mu_m$ denote the corresponding measure of their evolved fields at time $t + 1$. Define $\bar\mu = \frac{1}{M} \sum_{m=1}^M \mu_m$ as the mixture of input field distributions and $\overline{\Psi_{\#}\mu} = \frac{1}{M} \sum_{m=1}^M \Psi_{m\#}\mu_m$ as the corresponding mixture of output field measures. Under the same decomposition as in equation 4, the expected joint training error satisfies

$$\frac{1}{M} \sum_{m=1}^M \mathbb{E}_{x \sim \mu_m}\left[\|\Psi_m(x) - F_\theta(x)\|_2^2\right] \lesssim \varepsilon_{\text{approx}}^2 + R^{\bar\mu}(V_x) + R^{\overline{\Psi_{\#}\mu}}(V_y) + \mathcal{O}\left(\frac{1}{\sqrt{N}}\right). \tag{7}$$

Here, the projection terms depend on the shared mixture distributions rather than on any single PDE family. By exposing the network to diverse yet structurally related dynamics, multi-PDE pretraining enables the residual encoder, bottleneck, and decoder to learn common multiscale feature subspaces that align with recurring spatial and spectral patterns across PDE families. This shared representation captures a greater portion of the underlying functional variability with the same latent dimensionality, thereby reducing the average projection error. As a result, **during downstream finetuning**, the pretrained residual U-Net requires only minor residual adaptations in its local blocks,

leading to improved numerical stability, and strong cross-family generalization without increasing parameter count.

### 4.3 PRETRAINING AND FINETUNING DATASET

SPUS is pretrained on a diverse set of PDE types from the PDEGYM dataset (Herde et al., 2024), which includes four operators derived from the compressible Euler (CE) equations:

- **CE-RP**, containing trajectories initialized with four-quadrant Riemann problems;
- **CE-CRP**, initialized with multiple curved Riemann problems;
- **CE-KH**, representing shear-driven Kelvin–Helmholtz instabilities; and
- **CE-Gauss**, featuring initial conditions with Gaussian vorticity profiles.

Each dataset consists of 10,000 trajectories. Each trajectory has 21 time steps and each time step consists of five physical fields: density $\rho$, horizontal velocity $u$, vertical velocity $v$, pressure $p$, and energy $E$ with spatial grid of resolution $128 \times 128$.

We fine-tune SPUS on seven previously unseen downstream PDEs from the PDEGYM dataset, using 128 trajectories for each PDE task. These downstream PDEs include three operators governed by the CE equations, three operators governed by the incompressible Navier-Stokes (NS) equations, and one based on the wave equation:

- **CE-RPUI:** consisting of trajectories initialized with four-quadrant Riemann problems featuring uncertain interfaces;
- **CE-RM:** representing the Richtmyer-Meshkov instability problem;
- **NS-PwC:** initialized from piecewise-constant vorticity fields;
- **NS-SL:** initialized with double shear layer conditions;
- **FNS-KF:** also initialized from piecewise-constant vorticity fields;
- **Wave-Gauss:** containing trajectories initialized as a sum of Gaussians that are propagated by the spatially varying wave speed.
- **SE-AF:** contains contains the steady-state density over airfoils

Each CE-RPUI and CE-RM trajectory contains 21 time steps. Each time step has five physical fields: density $\rho$, horizontal velocity $u$, vertical velocity $v$, pressure $p$, and energy $E$. On the other hand, each trajectory in the three NS datasets also has 21 time steps but only two physical fields: horizontal velocity $u$, and vertical velocity $v$. For the Wave-Gauss dataset, trajectories have 15 time steps with one physical field, spatially varying wave speed $w$. For the SE-AF dataset, the samples are time-independent and solution operator maps a shape coefficient into the steady state solution. All fine-tuning datasets share a common spatial resolution of $128 \times 128$ grid points.

### 4.4 FINETUNING STRATEGIES AND BASELINE MODELS

In downstream tasks, the number of variables per time step may differ from those used during pretraining. To adapt the pretrained model to downstream tasks with different input and output dimensions than pretraining, we introduce lightweight input and output adapters. Specifically, we use $1 \times 1$ convolutional layers as adapters:

- The **InputAdapter** maps the task-specific input (e.g., 2 fields for NS-SL) to the 5-field format expected by the pretrained SPUS model.
- The **OutputAdapter** maps the model's 5-field output back to the task-specific output dimensionality (e.g., 2 fields for NS-SL).

These adapters are simple, efficient, and allow the pretrained model to be flexibly applied to a variety of downstream tasks without modifying its internal architecture. For each downstream task, we fine-tuned either the pretrained model or the pretrained model with adapters (if the number of the fields differed from five) using 128 trajectories. The model was fine-tuned for 200 epochs using

Table 1: Comparison of model performance (average MSE over all predicted timesteps from the initial conditions of the trajectories) on six unseen downstream PDE datasets fine-tuned with 128 trajectories. Lower is better. The U-Net* is trained from scratch using 128 trajectories for each downstream PDE dataset.

| Dataset | SPUS (Ours, 36M) | DPOT (122M) | POSEIDON (158M) | U-Net* (36M) |
|---|---|---|---|---|
| CE-RPUI | **0.0054** | 0.0570 | 0.0085 | 0.0337 |
| CE-RM | **0.0159** | 0.0222 | 0.4181 | 0.0218 |
| NS-PwC | 0.0048 | 0.0294 | **0.0004** | 0.0048 |
| FNS-KF | **0.0015** | 0.0301 | 0.0017 | 0.0047 |
| NS-SL | **0.0163** | 0.1461 | **0.0163** | 0.0165 |
| SE-AF | **0.0006** | - | 0.0031 | 0.0040 |
| Wave-Gauss | 0.0069 | 0.0107 | **0.0068** | 0.0071 |

the Adam optimizer with a linear learning rate schedule starting from $10^{-4}$, and a batch size of 10. The learning rate decreased linearly over the course of training.

To ensure a fair comparison, we fine-tune two baseline FMs: DPOT "M" (122M parameters) (Hao et al., 2024) and POSEIDON "B" (158M parameters) (Herde et al., 2024). DPOT was pretrained on 12 PDE datasets governed by the Navier-Stokes, diffusion-reaction, and shallow-water equations, whereas POSEIDON was pretrained on 6 PDE datasets governed by the compressible Euler and Navier-Stokes equations. For both baselines, we adopt the exact hyperparameter settings recommended in their original papers and accompanying code repositories (Hao et al., 2024; Herde et al., 2024). All models, including SPUS, are fine-tuned separately on each downstream PDE task using the same set of 128 trajectories for 200 epochs with MSE loss. Performance is evaluated on testing dataset corresponding to each PDE task.

DPOT recommends a context window of 10 timesteps. Accordingly, to predict trajectories from their initial conditions, we follow the same fine-tuning methodology described in Herde et al. (2024), padding input sequences with timestep 0 when predicting steps earlier than the 10th. For instance, to predict the state at timestep 4, the input sequence is padded as follows:

$$[\mathrm{ts}_0, \mathrm{ts}_0, \mathrm{ts}_0, \mathrm{ts}_0, \mathrm{ts}_0, \mathrm{ts}_0, \mathrm{ts}_0, \mathrm{ts}_1, \mathrm{ts}_2, \mathrm{ts}_3].$$

POSEIDON, on the other hand, is designed to take a single timestep as input, along with the corresponding $\Delta t$, and directly predict any future frame within the trajectory. This allows POSEIDON to predict any timestep (using only the initial timestep as context) without requiring an autoregressive rollout. In practice, such "direct" predictions result in higher average accuracy compared to predictions generated via autoregressive rollout. Therefore, we report POSEIDON's performance based on its direct prediction accuracy.

To further assess the parameter efficiency and architectural simplicity of SPUS, we trained a U-Net model with 36M parameters—sharing the same architecture as SPUS—from scratch using the same set of 128 trajectories per downstream dataset on which SPUS, POSEIDON, and DPOT were fine-tuned, and compared its performance with these foundation models.

The comparison of model performance—measured as average mean squared error (MSE) across all predicted timesteps from the initial condition of the trajectories—on seven unseen downstream PDE datasets fine-tuned with 128 trajectories is presented in Table 1.

## 5 EXPERIMENTS

**Is SPUS an effective lightweight PDE emulator? Does SPUS with only 36 million parameters generalize as accurately as larger models?** To address these questions, we design and evaluate the following three experiments.

**(A). Does SPUS generalize to unseen systems governed by the compressible Euler (CE) equations, consistent with its pretraining?** We investigate whether SPUS can generalize to previously

unseen physical systems that are governed by CE equations, consistent with its pretraining. To evaluate this, we fine-tune the pretrained SPUS model on the CE-RPUI dataset. While this dataset adheres to the same underlying physical laws, its distribution of initial conditions differs from those seen during pretraining, presenting a clear out-of-distribution (OOD) generalization challenge (Herde et al., 2024). As shown in Table 1, SPUS achieves strong performance in autoregressively predicting full trajectories from initial conditions, despite having only 36 million parameters. Notably, it outperforms both the substantially larger POSEIDON model (158 million parameters) and the DPOT model (122 million parameters) in terms of average mean squared error (MSE) across 240 test trajectories. A randomly selected trajectory prediction from the CE-RPUI test set is shown in Figure A.2, where the SPUS predictions closely match the ground truth at each time step. These results demonstrate the effectiveness and computational efficiency of the lightweight SPUS model relative to significantly larger architectures.

We also fine-tune the pretrained SPUS model on the CE-RM dataset, which exhibits significantly more complex dynamics compared to CE-RPUI. SPUS demonstrates strong generalization capability in predicting entire trajectories from initial conditions, as illustrated in Figure A.3. Furthermore, as shown in Table 1, SPUS achieves a lower average MSE across 130 test trajectories compared to both the POSEIDON and DPOT models, despite their substantially larger parameter counts.

**(B). Does SPUS generalize to systems governed by equations different from those used in pretraining?** We investigate the ability of SPUS to generalize to previously unseen physical systems governed by equations different from those used during pretraining. Specifically, we fine-tune the pretrained SPUS model on three datasets governed by incompressible NS equations that were not part of the pretraining data: NS-PwC, NS-SL, and FNS-KF. As shown in Table 1, despite not being exposed to incompressible NS dynamics during pretraining, surprisingly, SPUS achieves superior time-step prediction performance compared to DPOT across all three datasets—even though DPOT was pretrained on operators of both compressible and incompressible NS equations. For the POSEIDON model, whose pretraining data includes two operators governed by NS equations, SPUS outperforms it on FNS-KF, matches its performance on NS-SL, and is outperformed on NS-PwC, as summarized in Table 1. These results demonstrate the strong generalization capability of SPUS to new physical regimes outside its pretraining distribution and highlight its effective transferability to downstream tasks governed by equations different from those seen during pretraining. Randomly selected trajectory predictions from the test datasets of NS-PwC, NS-SL, and FNS-KF are shown in Figure A.4 and Figure A.5 (in Appendix). As observed, SPUS demonstrates strong generalization performance on each of the NS dataset; however, the predicted variables gradually deviate from the ground truth over time. We also fine-tuned SPUS on the Wave-Gauss dataset, which is governed by the wave equation. As shown in Table 1, SPUS outperforms DPOT and is narrowly outperformed by POSEIDON for 240 Wave-Gauss testing trajectories.

**(C). Does SPUS generalize to time-independent PDEs although it is pretrained on time-dependent PDEs?** We fine-tune the pretrained SPUS model on the time-independent SE-AF trajectories to evaluate its generalization capability beyond time-dependent PDEs. Although SPUS is originally pretrained on time-dependent PDEs, it can be readily adapted to time-independent problems because it does not explicitly take time as an input variable. Instead, the model simply learns the mapping between input and output fields required for prediction in static PDEs. As shown in Table 1, SPUS achieves better performance than both POSEIDON and the unpretrained U-Net on the SE-AF testing dataset. A randomly selected prediction from the SE-AF test set is illustrated in Figure A.7.

**Does SPUS show scalability with dataset size?** Table 4 reports the average MSE of predicting the time steps of entire trajectories from the initial condition of the trajectories across six downstream PDEs using SPUS, finetuned with 32, 128, and 256 trajectories. As shown, increasing the fine-tuning set size reduces the MSE across the six downstream PDEs, demonstrating that SPUS scales favorably with additional data.

**Summaries of the experiments** Based on the above experiments, we observe that SPUS, built on a residual U-Net architecture with only 36 million parameters, achieves state-of-the-art generalization on downstream tasks, outperforming significantly larger models such as POSEIDON (158 million parameters) and DPOT (122 million parameters). These results highlight that a simple, well-

established architecture—specifically, a residual U-Net—can be effectively leveraged as a foundation model (FM) for PDEs. Despite its architectural simplicity and relatively small parameter count, SPUS is capable of capturing complex dynamics and performs competitively with more sophisticated, larger models. We also observe that SPUS, when pretrained on a diverse set of simpler PDEs (such as CE), demonstrates strong performance on complex downstream PDEs (such as NS). This indicates the effective transferability of SPUS across distinct physical regimes, despite differences in the underlying governing equations. Furthermore, this suggests that even when the pretraining data are derived from PDEs governed by simple CE equations, a sufficiently diverse pretraining dataset—spanning variations in initial and boundary conditions, domain geometries, and external forcing—can enable the FM to generalize effectively. Moreover, SPUS is pretrained to emulate the behavior of numerical solvers by autoregressively predicting the next time step from the current one. The results on downstream tasks for SPUS suggest that this pretraining strategy helps the FM learn the underlying physics of PDEs, enabling more accurate and physically consistent predictions.

Additional results on the performance evaluation of SPUS on downstream tasks, as well as visual comparisons of SPUS's performance on trajectory prediction from initial conditions with larger models (POSEIDON and DPOT), are presented in Appendices A.1–A.6.

## 6    LIMITATIONS

SPUS is pretrained on a limited set of PDE families, using only four CE datasets. Despite this narrow pretraining scope, SPUS shows promising transferability across equation types, generalizing from compressible Euler to incompressible Navier–Stokes and wave equations. Currently, SPUS cannot simultaneously predict multiple future timesteps from an initial condition. In future work, we plan to extend SPUS to support both direct and autoregressive temporal prediction. We also expect that pretraining on a broader range of governing equations would further enhance its generalization capability. Moreover, the current architecture is restricted to regular geometries, and extending SPUS to irregular domains remains an important direction for future research.

## 7    CONCLUSIONS

We propose SPUS, a compact and lightweight FM for PDEs, capable of handling a broad range of physical systems. The model is based on a simple residual U-Net architecture and is trained using a straightforward autoregressive pretraining strategy. Despite its relatively small size—only 36 million parameters—SPUS demonstrates strong generalization capabilities across six diverse downstream PDE tasks. SPUS consistently outperforms the significantly larger DPOT model across all downstream datasets. When compared to the POSEIDON model, which also has substantially more parameters, SPUS achieves superior performance on three datasets, matches performance on one, is narrowly outperformed on another (MSE: 0.0069 vs. 0.0068), and is outperformed on one task. These results establish SPUS as a highly parameter-efficient foundation model, capable of solving a wide range of complex PDE systems with competitive accuracy. Furthermore, we demonstrate that pretraining SPUS on simpler PDEs (such as CE) with autoregressive training to emulate a numerical solver enables effective transfer to more complex PDEs (such as NS), reducing the amount of data required for finetuning even when the downstream task involves more complex dynamics than those seen during pretraining.

## REPRODUCIBILITY STATEMENT

All datasets used in this work are publicly available. The code will be released at the time of publication.

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

# A APPENDIX

## A.1 ERROR GROWTH OVER TIME

Figure A.1 presents the average mean squared error (MSE) of trajectory predictions over time for the FNS-KF test datasets. As shown, the SPUS model—with only 36 million parameters—exhibits an approximately linear increase in prediction error over time, a behavior consistently observed across our downstream datasets. POSEIDON, a larger FM with 158 million parameters, demonstrates a similar error growth pattern as shown in Figure A.1b. As can be seen, SPUS achieves comparable performance to POSEIDON (Direct) at early time steps and surpasses it at later time steps. These results highlight the potential of SPUS to deliver accurate long-term predictions despite having significantly fewer parameters.

Table 2: Comparison of model performance (average MSE across all predicted 20 timesteps from the initial conditions of the trajectories) on the CE-RM, NS-SL, and FNS-KF PDE datasets, fine-tuned with 128 trajectories using three different pretrained model sizes. SPUS demonstrates scalability with increasing model size.

| Dataset | SPUS-9M | SPUS-36M | SPUS-76M |
|---------|---------|----------|----------|
| CE-RM   | 0.0203  | 0.0159   | 0.0129   |
| NS-SL   | 0.0247  | 0.0163   | 0.0027   |
| FNS-KF  | 0.0045  | 0.0015   | 0.0008   |

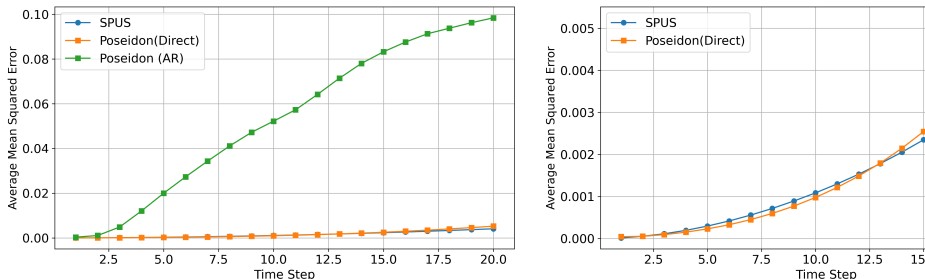

(a) Prediction error over time for FNS-KF with Poseidon (Direct), Poseidon (AR), and SPUS

(b) Prediction error over time for FNS-KF with Poseidon (Direct), and SPUS

Figure A.1: Average MSE of trajectory predictions over time for the FNS-KF test datasets. SPUS (36M) shows approximately linear error growth similar to POSEIDON (Direct). It surpasses POSEIDON (direct) at later steps on FNS-KF, highlighting its efficiency and long-term accuracy as shown in Figure (b).

Table 3: Performance analysis of SPUS with different numbers of residual blocks at each stage. The table reports the average MSE across all 20 predicted timesteps from the initial timestep on the CE-RM, NS-SL, and FNS-KF test datasets. Using five residual blocks per stage in SPUS reduces the prediction MSE across all downstream tasks.

| Model | Number of Residual Blocks | CE-RM | NS-SL | FNS-KF |
|-------|---------------------------|-------|-------|--------|
| SPUS-36M | 2 | 0.0159 | 0.0163 | 0.0015 |
| SPUS-76M | 5 | 0.0129 | 0.0027 | 0.0008 |

## A.2 SPUS SCALES WITH MODEL SIZE

We pretrain SPUS with three model sizes to study scalability:

- **SPUS-9M:** This variant contains approximately 9 million parameters. Its architecture is identical to SPUS-36M except that the number of channels in each encoder and decoder stage is reduced by half.

Table 4: Evaluation (average MSE on all predicted timesteps from initial condition) of SPUS on downstream datasets under different numbers of finetuned trajectories

| Downstream Dataset | Number of Trajectories | | |
|:---:|:---:|:---:|:---:|
| | 32 | 128 | 256 |
| CE-RPUI | 0.0057 | 0.0054 | 0.0041 |
| CE-RM | 0.0246 | 0.0159 | 0.0130 |
| NS-PwC | 0.0076 | 0.0048 | 0.0025 |
| NS-SL | 0.0286 | 0.0163 | 0.0004 |
| FNS-KF | 0.0098 | 0.0015 | 0.0012 |
| Wave-Gauss | 0.0097 | 0.0069 | 0.0068 |

- **SPUS-36M:** The baseline configuration illustrated in Figure 2. It includes two residual blocks at each level of the encoder, decoder, and bottleneck.

- **SPUS-76M:** This larger variant follows the same architecture as SPUS-36M but increases the number of residual blocks in each encoder, decoder, and bottleneck stage from two to five following Huang et al. (2023) .

All three pretrained models are fine-tuned on three downstream datasets: CE-RM, NS-SL, and FNS-KF. Table 2 summarizes the average test MSE across all predicted timesteps for each dataset and model configuration. As shown, increasing model size consistently reduces the test error across for all three downstream datasets, demonstrating that SPUS scales effectively with model size.

### A.3 PERFORMANCE EVALUATION OF SPUS WITH DIFFERENT NUMBERS OF RESIDUAL BLOCKS PER STAGE

As shown in Table 3, increasing the number of residual blocks in each stage of SPUS improves model accuracy. Specifically, the larger variant, SPUS-76M with five residual blocks per encoder, decoder, and bottleneck stage, achieves lower prediction errors on all three downstream PDE datasets compared to the configuration with two residual blocks (SPUS-36M), demonstrating the benefit of deeper residual refinement within each stage.

### A.4 PERFORMANCE EVALUATION OF SPUS WITH AUTOREGRESSIVE TRAINING UNDER A FIRST-ORDER MARKOV ASSUMPTION

We formulate the autoregressive training of SPUS as a first-order Markov process, where the evolution of the system depends only on its immediately preceding state. This formulation contrasts with DPOT, which conditions on the previous ten timesteps to predict the next one. Relying on multiple past states, as in DPOT, to start inference, it requires the 10 past states to be generated from expensive numerical simulator. On the other hand, SPUS only need the initial condition which has no computational cost to start the inference.

Relying on multiple past states, as in DPOT, can accumulate redundant temporal information and potentially introduce compounding errors during long rollouts. As shown in Table 1, our first-order Markov formulation enables SPUS to achieve lower prediction errors and improved long-term stability across all downstream datasets compared to DPOT.

### A.5 PERFORMANCE EVALUATION OF SPUS ON THE CE-RPUI, CE-RM, FNS-KF AND WAVE-GAUSS DATASETS

Figure A.2 presents randomly selected trajectory prediction from the CE-RPUI test dataset. Notably, the predicted variables closely match the ground truth at each time step, although the deviation between prediction and ground truth increases more noticeably over time for CE-RM compared to CE-RPUI due to its more complex dynamics.

Figure A.3 presents randomly selected trajectory prediction from the CE-RM test dataset. Notably, the predicted variables closely match the ground truth at each time step.

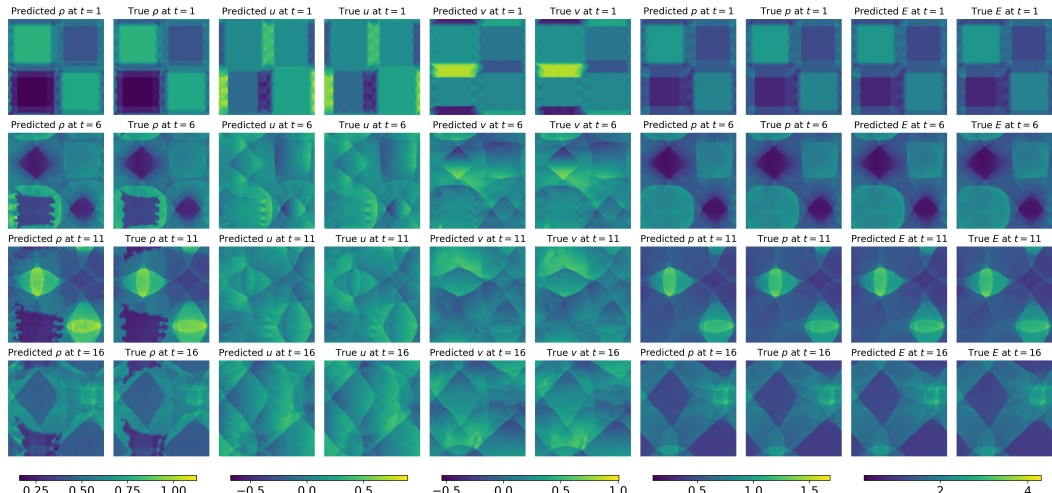

Figure A.2: Autoregressive trajectory prediction by SPUS from the initial condition of a randomly selected trajectory in the CE-RPUI testing dataset (240 test trajectories). The figure shows example results at time steps $t = 1, 6, 11, 16$ for five system variables: density $\rho$, horizontal velocity $u$, vertical velocity $v$, pressure $p$, and energy $E$. SPUS takes the initial condition $u'_t = u_{t=0}$ as input and recursively predicts subsequent states based on its own previous outputs for $t = 1, \ldots, 20$, as described in Figure 1 (inference step). As shown, the predicted variables closely match the ground truth at each time step.

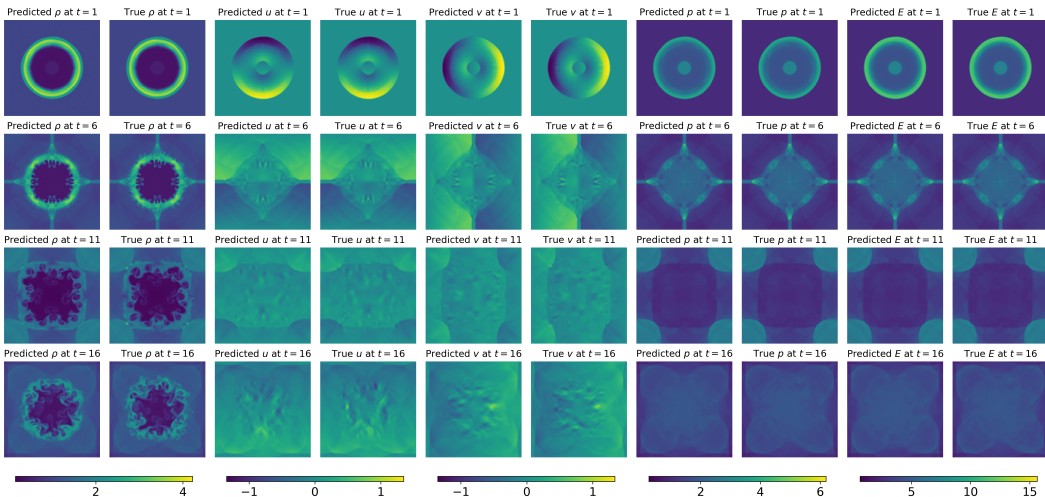

Figure A.3: Autoregressive trajectory prediction by SPUS from the initial condition of a randomly selected trajectory in the CE-RM testing dataset (130 test trajectories). The figure shows example results at time steps $t = 1, 6, 11, 16$ for five system variables: density $\rho$, horizontal velocity $u$, vertical velocity $v$, pressure $p$, and energy $E$. SPUS takes the initial condition $u'_t = u_{t=0}$ as input and recursively predicts subsequent states based on its own previous outputs for $t = 1, \ldots, 20$, as described in Figure 1 (inference step). As shown, the predicted variables closely match the ground truth at each time step, although the deviation between prediction and ground truth increases more noticeably over time for CE-RM compared to CE-RPUI due to its more complex dynamics.

Figure A.4 presents randomly selected trajectory prediction from the NS-PwC and NS-SL test dataset. Notably, although SPUS was not pretrained on incompressible Navier–Stokes dynamics, its predictions closely follow the ground truth variables at each time step, demonstrating robust generalization to the NS-PwC and NS-SL system.

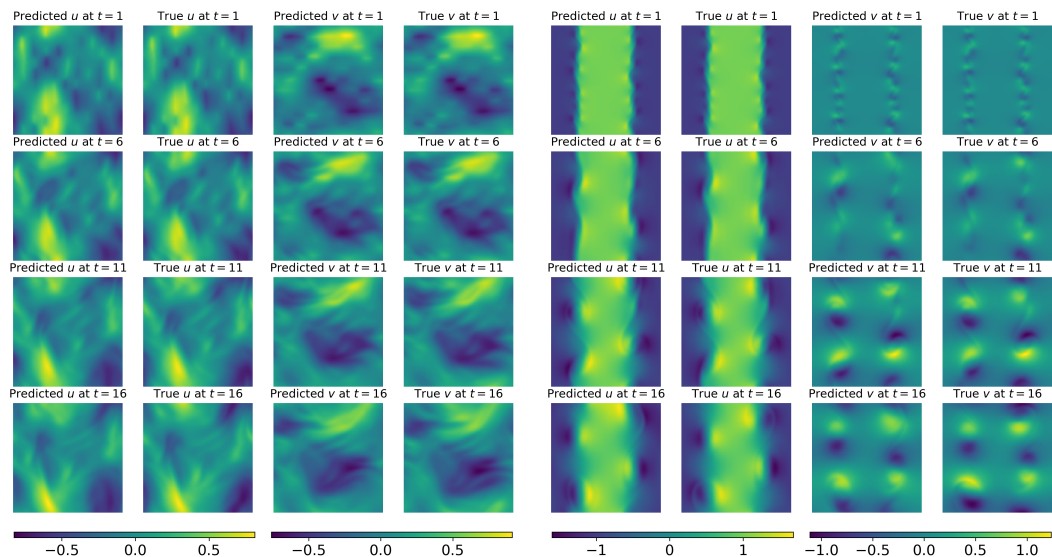

(a) Autoregressive trajectory prediction from initial state for NS-PwC

(b) Autoregressive trajectory prediction from initial state for NS-SL

Figure A.4: Autoregressive trajectory prediction by SPUS from the initial condition of a randomly selected trajectory in the NS-PwC and NS-SL testing datasets (each with 240 test trajectories). The figure shows example results at time steps $t = 1,\ 6,\ 11,\ 16$ for two system variables: horizontal velocity $u$, vertical velocity $v$. As shown, despite not being exposed to incompressible NS dynamics during pretraining, the predicted variables by SPUS closely match the ground truth at each time step. The deviation between prediction and ground truth (GT) increases more noticeably over time for NS-PwC compared to NS-SL.

Figure A.5 presents randomly selected trajectory prediction from the FNS-KF test dataset. Notably, although SPUS was not pretrained on incompressible Navier–Stokes dynamics, its predictions closely follow the ground truth variables at each time step, demonstrating robust generalization to the FNS-KF system.

Figure A.6 illustrates a representative trajectory prediction from the Wave-Gauss test set, demonstrating SPUS's ability to generalize to the Wave-Gauss system. However, in comparison to the Navier–Stokes downstream tasks, the deviation between the predicted variables and the ground truth increases more noticeably over time for Wave-Gauss.

### A.6 VISUALIZATION OF SPUS PERFORMANCE ON TRAJECTORY PREDICTION COMPARED WITH POSEIDON AND DPOT

Figure A.8 shows a random trajectory predictions for FNS-KF made by SPUS (36M), POSEIDON (158M), and DPOT (122M). Each model is finetuned with 128 trajectories.

Figure A.9 shows a random trajectory predictions for NS-SL made by SPUS (36M), POSEIDON (158M), and DPOT (122M). Each model is finetuned with 128 trajectories.

### A.7 CAN SPUS GENERALIZE TO DIFFERENT INPUT RESOLUTIONS?

We fine-tuned SPUS-36M using 128 PDE trajectories governed by the Burgers equation (Zhou & Farimani, 2024). Each trajectory consists of 100 time steps, with the velocity field represented on a spatial grid of size $64 \times 64$. For comparison, we also trained an unpretrained U-Net model with 36M parameters—sharing the same architecture as SPUS—from scratch, using the same 128 trajectories.

We evaluated both models on 64 test trajectories, predicting all 99 future time steps from the initial condition. The average mean squared error (MSE) across all 99 predicted time steps for the 64 trajectories was 0.0035 for SPUS and 0.0079 for the unpretrained U-Net. These results demonstrate

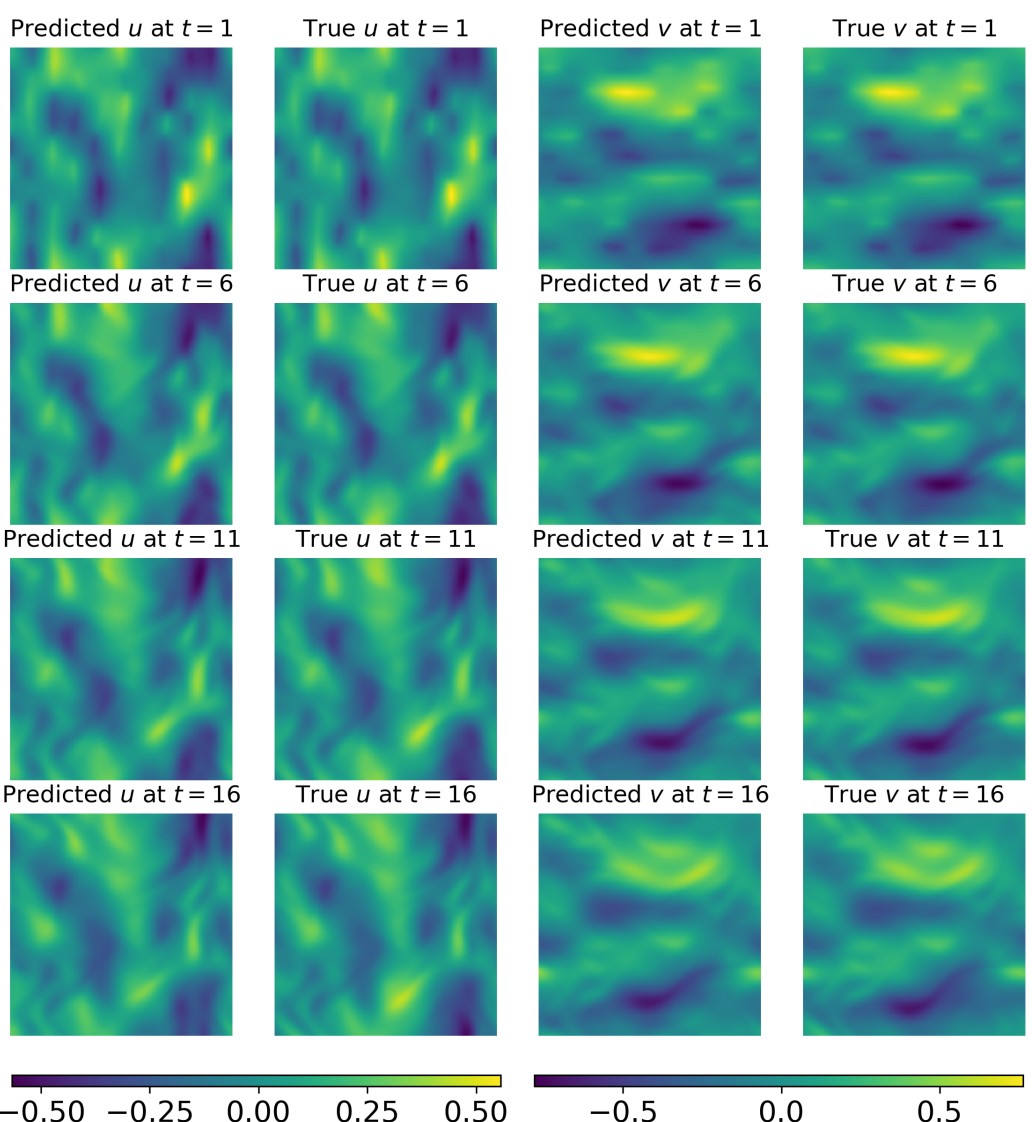

Figure A.5: Autoregressive trajectory prediction by SPUS from the initial condition of a randomly selected trajectory in the FNS-KF testing dataset (240 trajectories). The figure shows example results at time steps $t = 1, 6, 11, 16$ for two system variables: horizontal velocity $u$, vertical velocity $v$. SPUS takes the initial condition $u'_t = u_{t=0}$ as input and recursively predicts subsequent states based on its own previous outputs for $t = 1, \ldots, 20$, as described in Figure 1 (inference step). As shown, the predicted variables closely match the ground truth at each time step.

the strong generalization capability of SPUS to different spatial resolutions and its robustness for longer rollouts. Figure A.10 shows a randomly selected trajectory prediction for PDEs governed by the Burgers equation, generated by SPUS (36M) and an unpretrained U-Net (36M). The figure shows example results at time steps $t = 21, 31, 41, 51, 99$ for the velocity field on a $64 \times 64$ spatial grid. As shown, the unpretrained U-Net begins to deviate from the ground truth (GT) after $t = 41$, whereas SPUS remains close to the GT across all time steps, demonstrating its potential for longer rollouts. Figure A.11 shows the average MSE over 99 predicted time steps for 64 test trajectories of Burgers equation. As shown, SPUS (36M) maintains a nearly constant MSE after timestep 40, remaining stable through the final prediction. In contrast, the unpretrained U-Net (36M) exhibits a steadily increasing MSE.

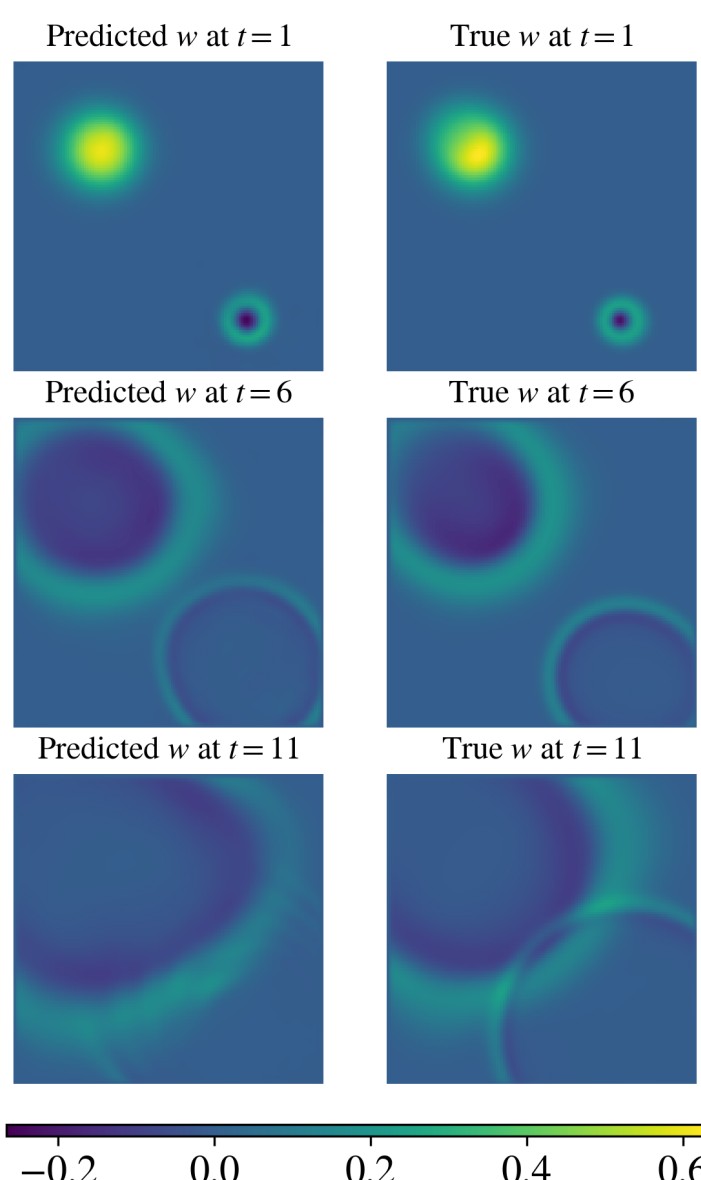

Figure A.6: Autoregressive trajectory prediction by SPUS from the initial condition of a randomly selected trajectory in the Wave-Gauss testing dataset (240 trajectories). The figure shows example results at time steps $t = 1, 6, 11$ for one system variable: wave speed $u$. SPUS takes the initial condition $u'_t = u_{t=0}$ as input and recursively predicts subsequent states based on its own previous outputs for $t = 1, \ldots, 14$, as described in Figure 1 (inference step). As shown, the predicted variables deviates very quickly from ground truth at each time step for Wave-Gauss compared to other downstream tasks.

### A.8 CAN SPUS SCALE TO PDES WITH HIGHER SPATIAL RESOLUTIONS THAN THOSE USED IN PRETRAINING?

SPUS is pretrained on PDEs with a spatial resolution of $128 \times 128$. To evaluate its scalability to higher-resolution PDEs, we finetuned the pretrained SPUS model using 128 trajectories governed by the Burgers equation (Zhou & Farimani, 2024). Each trajectory contains 20 time steps, with the velocity field represented on a $256 \times 256$ spatial grid. For comparison, we also trained a 36M-parameter U-Net—matching the architecture of SPUS—from scratch using the same 128 trajectories.

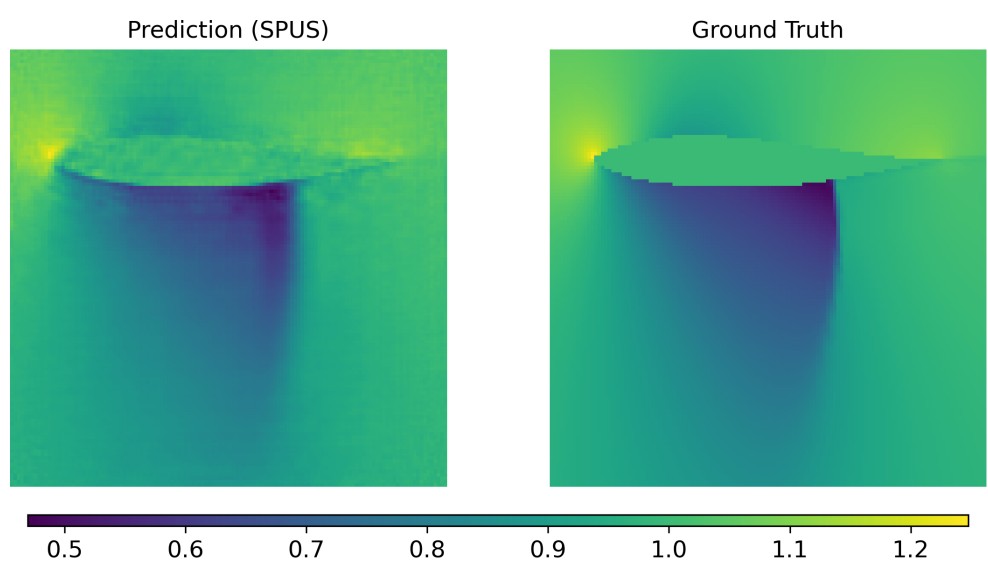

Figure A.7: A randomly selected predicted samples by SPUS for SE-AF

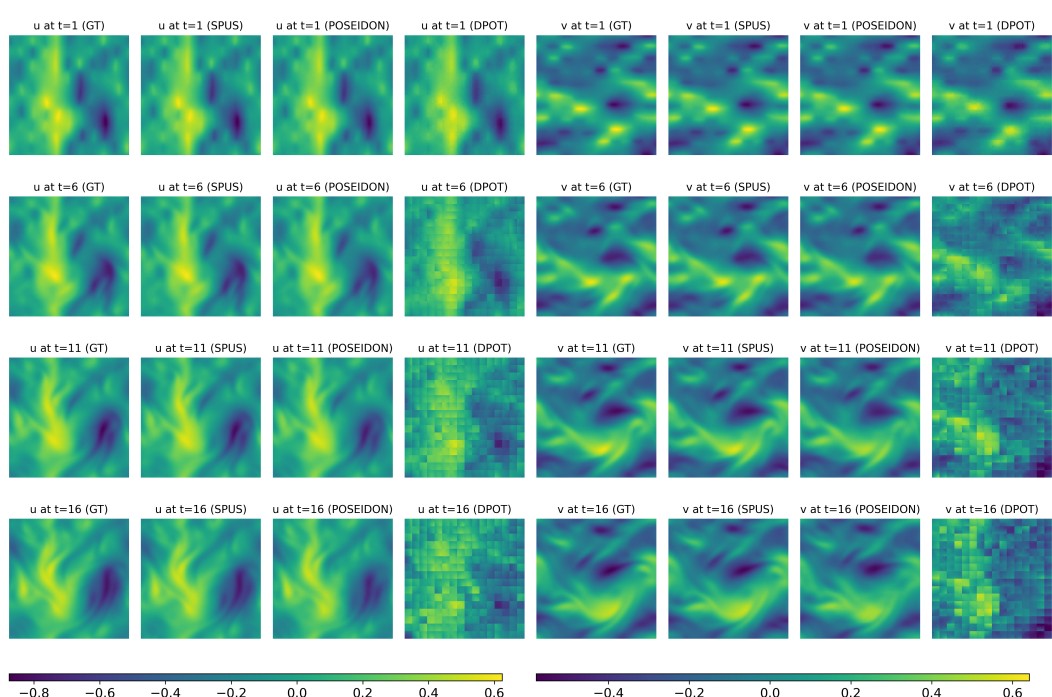

Figure A.8: A random trajectory predictions for FNS-KF made by SPUS (36M), POSEIDON (158M), and DPOT (122M). The figure shows example results at time steps $t = 1, 6, 11, 16$ for two system variables: horizontal velocity $u$, vertical velocity $v$.

We evaluated both models on 64 test trajectories, predicting all 19 future time steps from the initial condition. The average mean squared error (MSE) across the 19 predicted time steps was 0.0017 for SPUS and 0.0023 for the U-Net trained from scratch. These results demonstrate the strong generalization capability of SPUS to spatial resolutions higher than those used during pretraining.

Figure A.12 shows a randomly selected trajectory prediction for the Burgers equation generated by SPUS (36M) and the unpretrained U-Net (36M). Example velocity-field snapshots at time steps

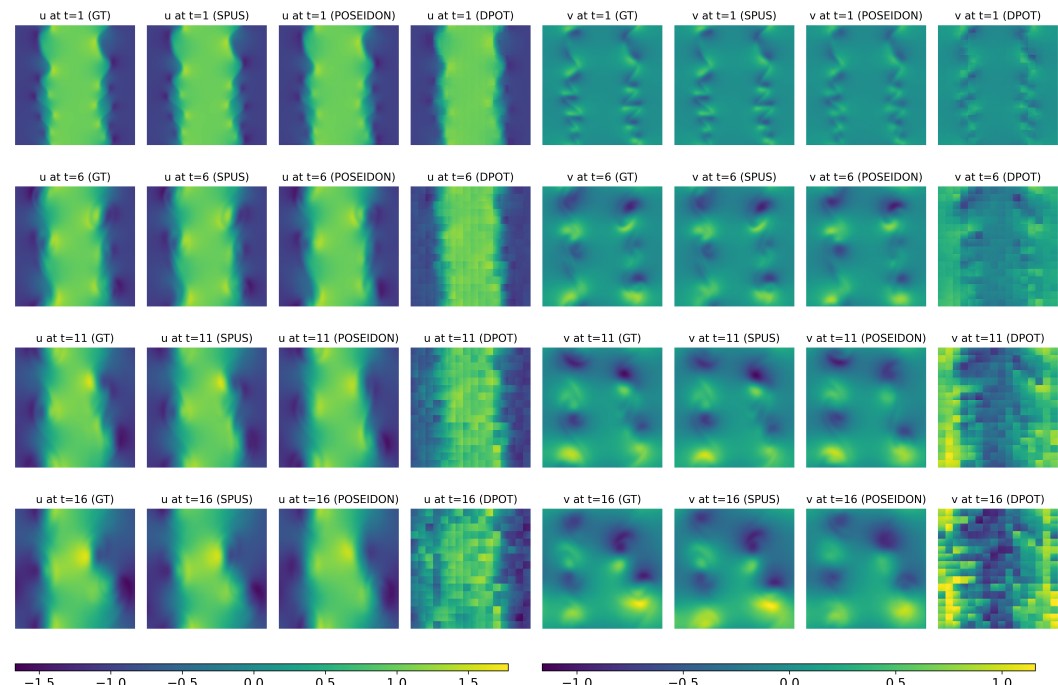

Figure A.9: A random trajectory predictions for NS-SL made by SPUS (36M), POSEIDON (158M), and DPOT (122M). The figure shows example results at time steps $t = 1, 6, 11, 16$ for two system variables: horizontal velocity $u$, vertical velocity $v$.

$t = 2, 6, 11, 16, 19$ on the $256 \times 256$ grid are presented. As illustrated, SPUS remains close to the ground truth across all time steps, whereas the U-Net trained from scratch develops noticeable periodic striping artifacts at later time steps that are absent in the ground-truth (GT) solution. These findings show that SPUS successfully scales to PDEs with higher spatial resolutions than those seen during pretraining.

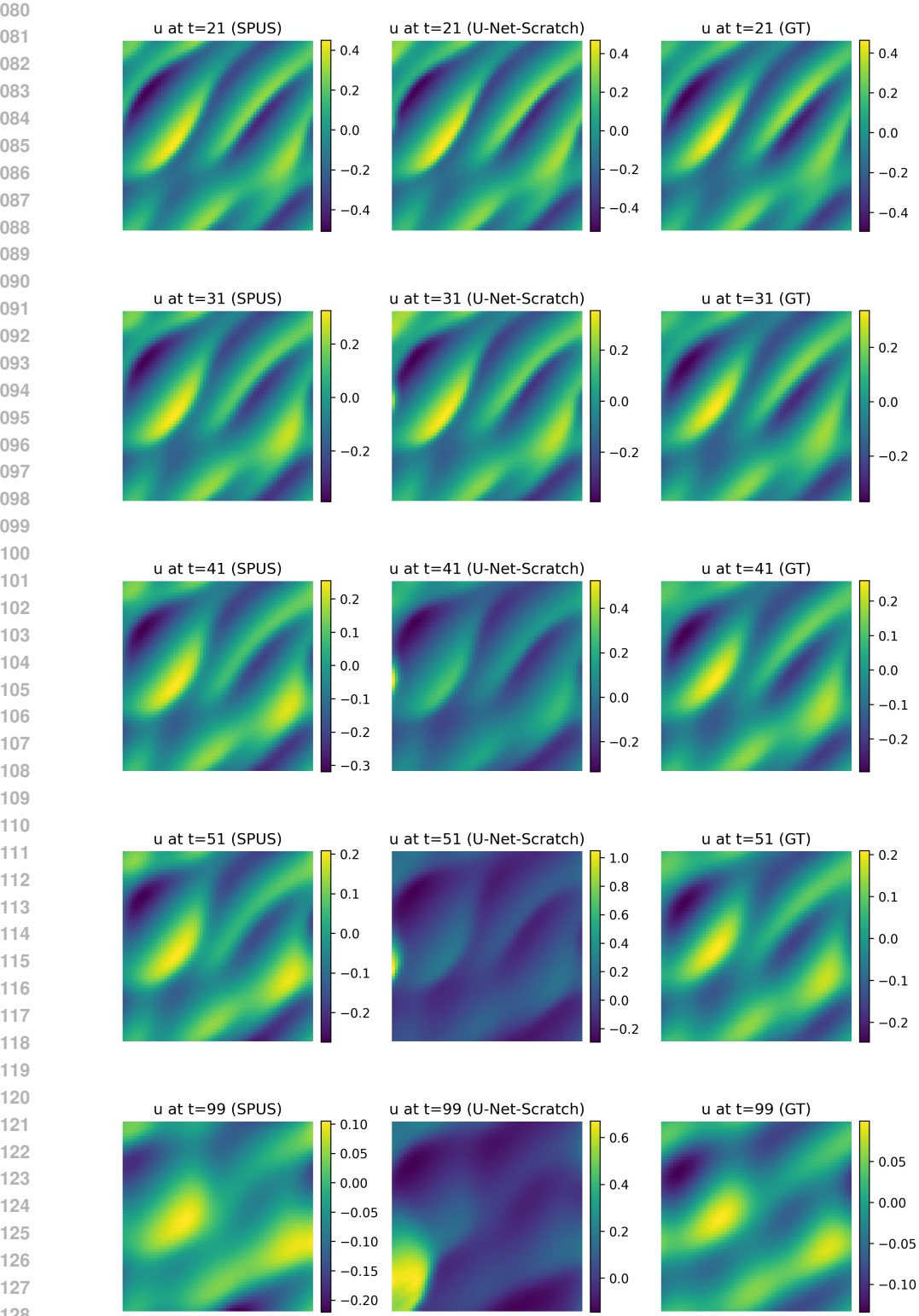

Figure A.10: A randomly selected trajectory prediction for PDEs governed by the Burgers equation, generated by SPUS (36M) and an unpretrained U-Net (36M). The figure shows example results at time steps $t = 21, 31, 41, 51, 99$ for the velocity field on a $64 \times 64$ spatial grid. As shown, the unpretrained U-Net begins to deviate from the ground truth (GT) after $t = 41$, whereas SPUS remains close to the GT across all time steps, demonstrating its potential for longer rollouts.

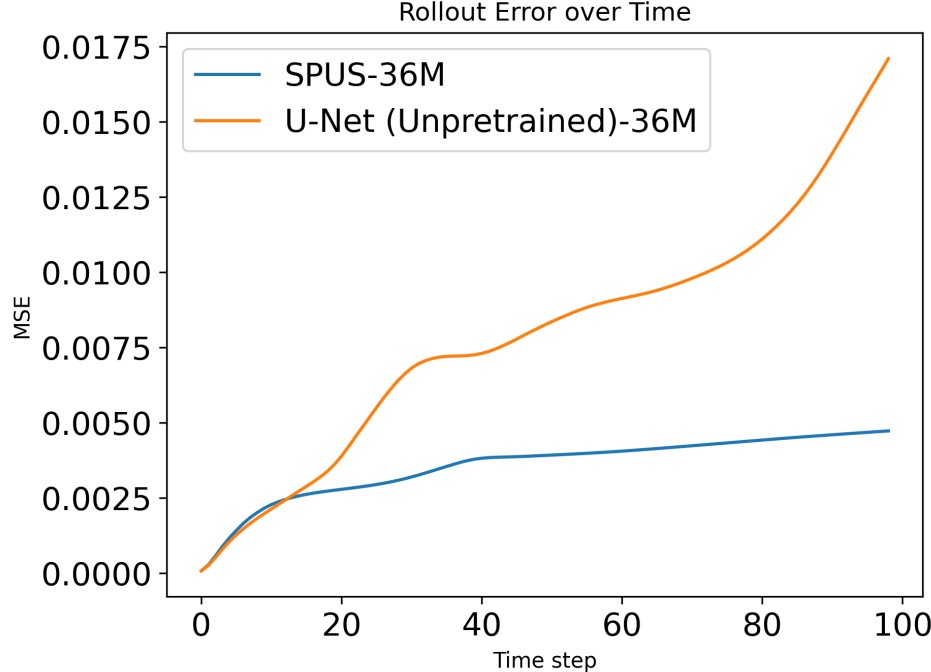

Figure A.11: Average MSE over 99 predicted time steps for 64 test trajectories of Burgers equation. As shown, SPUS (36M) maintains a nearly constant MSE after timestep 40, remaining stable through the final prediction. In contrast, the unpretrained U-Net (36M) exhibits a steadily increasing MSE.

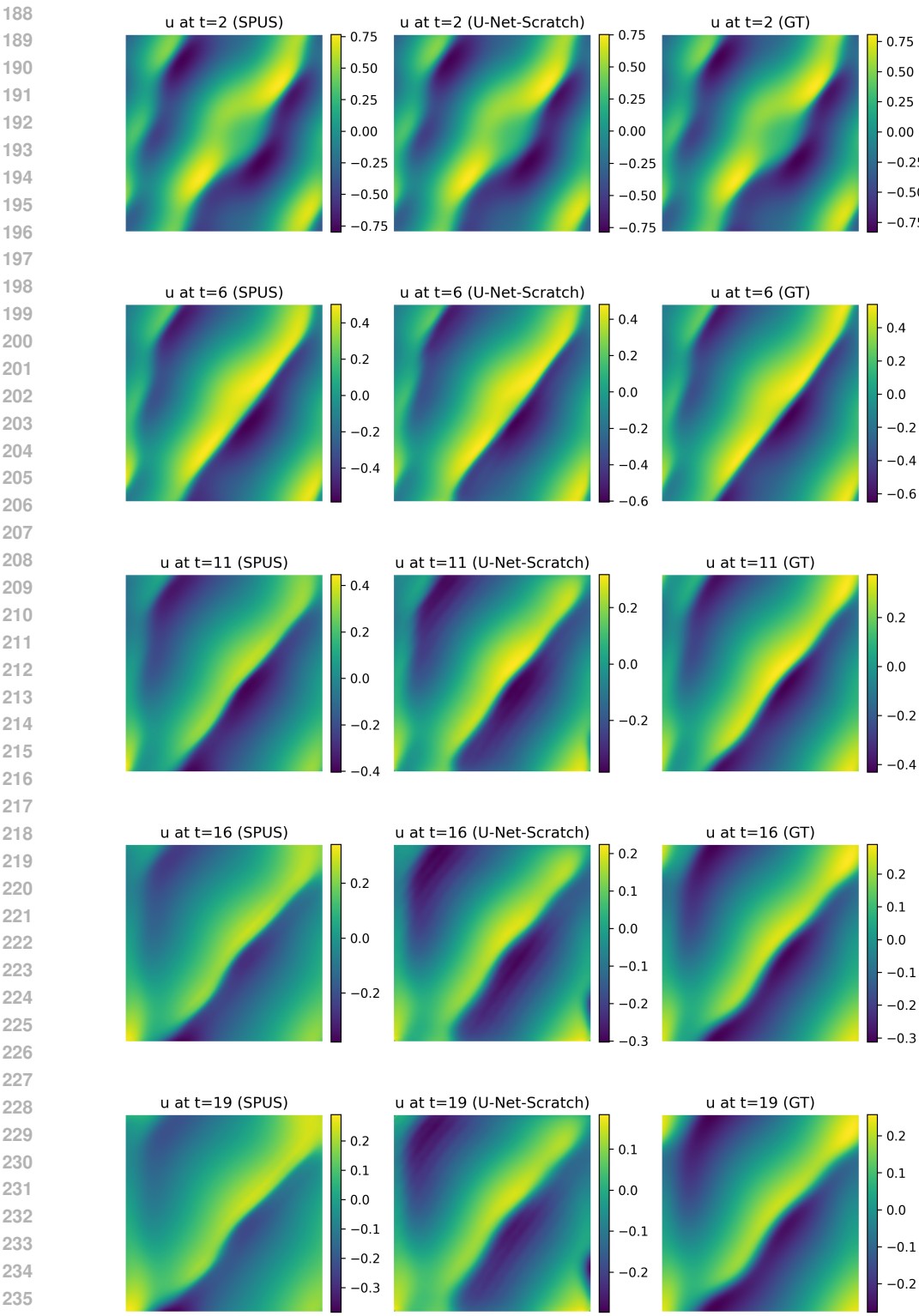

Figure A.12: Scalability of SPUS to PDE resolutions higher than those used in pretraining ($128 \times 128$). Shown is a randomly selected trajectory prediction for PDEs governed by the Burgers equation, generated by SPUS (36M) and an unpretrained U-Net (36M). The figure shows example results at time steps $t = 2, 6, 11, 16, 19$ for the velocity field on a $256 \times 256$ spatial grid. As illustrated, SPUS remains close to the ground truth across all time steps, whereas the U-Net trained from scratch develops noticeable periodic striping artifacts at later time steps that are absent in the ground-truth (GT) solution.

