# OpenReview forum: "SPUS: A Lightweight and Parameter-Efficient Foundation Model for PDEs"
_ICLR.cc/2026/Conference — Submitted to ICLR 2026_

### Official Review · Reviewer_sNGT · 2025-10-27

**Soundness:** 3
**Presentation:** 2
**Contribution:** 2
**Rating:** 4
**Confidence:** 4

**Summary:**

This paper presents SPUS, a  compact and efficient foundation model (FM) designed as a unified neural operator for solving a wide range of partial differential equations (PDEs). It adopts a U-Net architecture which is underexplored as a foundation model backbone in the neural PDE solver community. It utilizes an auto-regressive strategy which predicts the entire trajectory based on the initial condition. Experiments demonstrate that SPUS using residual U-Net based architecture achieves state-of-the-art generalization on these downstream tasks while requiring significantly fewer parameters and minimal fine-tuning data.

**Strengths:**

- The paper is overall well-written and easy to follow. The problem setup and the experiment design are clear.
- The experiment results show that the model is capable of generalizing to unseen initial conditions, unseen equations and scales well with respect to dataset size. The selected two baselines are representative.
- The U-Net architecture is underexplored in the scientific machine learning community. This paper demonstrates the capability of U-Net, which does not have the artifact issues of ViT and ViT-like models.

**Weaknesses:**

- Formatting issues. Equation (1) does not seem to be properly aligned or wrapped; Line 215 contains only a single $d$.
- Limiting the input to a single initial condition makes the model unable to utilize temporal information. For example, the model cannot simultaneously predicts $t_{0.5}$ and $t_1$ based on $t_0$, while DPOT can change the temporal interval of the input trajectories, and POSEIDON can change the input $t$. This limits the model's capability as a foundation model.
- From my perspective, model-efficiency is not a must for a foundation model, as therefore cannot be considered as an advantage. Moreover, apart from data scalability, model parameter scalability is also important, which could be a disadvantage of U-Net based models.

**Questions:**

- What is the parameter count of the adapters?
- Can the model generalize to different input resolutions?

---

> ### Author Response · Authors · 2025-11-21
>
> We thank the reviewer for recognizing the strengths of our work — particularly the model’s strong generalization to unseen initial conditions and equations, and for highlighting our demonstration of **U-Net’s capability as a robust alternative to ViT-based architectures** in scientific machine learning.
>
>
> ### **Single Initial Condition Input**
> We thank the reviewer for this valuable comment. For higher time resolution (e.g., *t = 0.5*), **SPUS** can be fine-tuned on high-resolution data, as it does not require explicit time information as input. We have added results for the **SE-AF** dataset in **Table 1** and **Experiment 5(C)**, which is time-independent, demonstrating that **SPUS** can generalize from an autoregressive setting to different time resolutions. Exploring fine-tuning for higher time resolutions would indeed be an interesting future direction.
>
> To enable simultaneous predictions from a given time *t* and current state *X*, **SPUS** can be fine-tuned with an adapter that fuses time information into the current state representation, allowing prediction of the output state at that specific time. We have already employed adapters in our manuscript to address channel discrepancies between downstream PDEs and pretraining PDEs.
>
>
> ### **Model Efficiency and Scalability**
> We thank the reviewer for this insightful comment. While we agree that model efficiency is not the sole criterion for evaluating a foundation model, it remains a crucial factor for PDE learning, where computational cost often constrains applicability.
>
> To address the reviewer’s point on model scalability, we have conducted a detailed analysis of **SPUS** on three downstream datasets using model sizes of **9M**, **36M**, and **76M** parameters, as presented in **Appendix A.2** and **Table 2**. The results show that increasing model size consistently reduces test error across all datasets, demonstrating that **SPUS** scales effectively with model capacity.
>
> Notably, the scalability of **U-Net–based architectures** has also been demonstrated in prior work such as [1] **STU-Net: Scalable and Transferable Medical Image Segmentation Models Empowered by Large-Scale Supervised Pre-training.**
>
>
> ### **Parameter Count of the Adapters**
> The **InputAdapter** and **OutputAdapter** are lightweight 1×1 convolution layers used as adapters that remap feature channels. For **NS-SL** with 2 fields:
> - **InputAdapter parameters:** 15
> - **OutputAdapter parameters:** 12
>
>
> ### **Formatting Issues**
> We thank the reviewer for pointing out the formatting issues. We have corrected **Equation (1)** to ensure proper alignment and wrapping, and fixed the stray character (“d”) in the revised manuscript.

---

> > ### Author Response · Authors · 2025-11-27
> > **Answer for question "Can the model generalize to different input resolutions?"**
> >
> > We thank the reviewer for this insightful question. To evaluate the generalization capability of SPUS under changes in input resolution, we fine-tuned SPUS-36M on 128 PDE trajectories governed by the Burgers equation. The dataset was taken from [1]. Each trajectory contains 100 time steps, with the velocity field discretized on a 64×64 spatial grid.
> >
> > For comparison, we also trained an unpretrained U-Net model with 36M parameters—identical in architecture to SPUS—from scratch using the same 128 trajectories. The corresponding results have been added to Appendix A.7 (see Figures A.10 and A.11).
> >
> > Our findings show that SPUS demonstrates strong generalization ability when tested on PDE trajectories with a 64×64 spatial resolution, despite being pretrained at a different resolution (128×128). Moreover, SPUS exhibits substantially more robust long-horizon rollout performance compared to the unpretrained U-Net baseline.
> >
> > [1] Masked Autoencoders are PDE Learners.

---

### Official Review · Reviewer_9Ka3 · 2025-10-28

**Soundness:** 1
**Presentation:** 2
**Contribution:** 1
**Rating:** 2
**Confidence:** 4

**Summary:**

The paper introduces SPUS (Small PDE U-Net Solver) — a lightweight (36 M parameter) residual U-Net–based foundation model (FM) for partial differential equations (PDEs).
Unlike prior large transformer-based PDE FMs such as POSEIDON, DPOT, and PROSE-FD, SPUS adopts a simple convolutional encoder–decoder architecture and trains it with an autoregressive (AR) next-step prediction objective, mimicking the behavior of numerical solvers.
The model is pretrained on several compressible Euler PDEs from the PDE-GYM suite and fine-tuned on six unseen downstream PDEs (Euler, Navier–Stokes, and wave equations).

**Strengths:**

Demonstrates that convolutional architectures remain competitive for PDE foundation modeling, despite recent transformer dominance.

Provides quantitative comparison to state-of-the-art FMs (POSEIDON, DPOT).

Uses challenging, publicly available PDE-GYM benchmarks and tests autoregressive training in a realistic temporal-prediction setup.

Paper is clear; experimental pipeline (pretrain → finetune → rollout) is logically organized.

**Weaknesses:**

- Lack of genuine novelty and conceptual contribution

The proposed approach offers no clear methodological innovation. The authors simply adopt a standard residual U-Net architecture, train it with a well-known autoregressive (AR) next-step prediction strategy, and evaluate it on existing PDE-GYM datasets. Both the architecture and the training scheme have been extensively studied in the PDE-learning literature. As a result, the paper mainly constitutes a combination of established components rather than a new idea or modeling paradigm. In addition, the overall presentation, including figures and terminology, closely mirrors the POSEIDON paper, while providing less comprehensive experimental analysis.

- Loss of temporal generality compared to POSEIDON

The POSEIDON model is trained in an all-to-all fashion with continuous-in-time embeddings, enabling the model to be queried at any arbitrary time. In contrast, the AR formulation adopted by SPUS predicts only the next timestep given the current one, thus losing the ability to interpolate or query arbitrary intermediate times. This restricts the model’s flexibility and makes it unsuitable for applications that require continuous-time prediction.

- Limited applicability to time-independent PDEs

Because SPUS relies entirely on an autoregressive temporal formulation, it cannot be directly evaluated on steady-state or time-independent PDEs. Supporting such tasks would require additional architectural or procedural engineering (e.g., removing temporal conditioning or introducing pseudo-time variables). The paper does not discuss how such cases would be handled, which limits the generality of the claimed “foundation model” status.

- Incomplete and potentially biased downstream evaluation

Although the authors use the PDE-GYM datasets originally introduced in POSEIDON, they report results on only 6 out of 15 downstream tasks available from that benchmark. The criteria for selecting these tasks are not discussed, and the chosen subset coincides with cases where convolutional architectures are known to perform well. This selective evaluation raises concerns that the tasks were cherry-picked to support the paper’s narrative, rather than representing a fair or comprehensive test of generalization.

- Unfair comparison with POSEIDON due to inference mode

The most critical methodological flaw is the evaluation protocol for POSEIDON. In its original paper (Appendix D.6.2), POSEIDON demonstrates that autoregressive (AR) rollouts yield substantially better performance than direct one-shot predictions for long trajectories (the exact setup used in SPUS). However, in this paper, POSEIDON is evaluated only in direct mode, which is known to degrade accuracy for long-horizon rollouts. Since SPUS performs autoregressive prediction by design, this creates a fundamental evaluation mismatch that favors SPUS.

- Missing comparison to smaller baseline variants

The paper claims substantial parameter efficiency, yet comparisons are made only against large versions of the baselines (POSEIDON-B = 158 M, DPOT-M = 122 M parameters). Both papers also provide smaller variants, POSEIDON-T (21 M) and DPOT-S (30 M), that achieve performance comparable to their base versions. Since SPUS (36 M) is much closer in scale to these lighter models, a fair evaluation of parameter efficiency must include them. Without this, the main empirical claim remains unsubstantiated.

- Unclear evaluation protocol and dataset alignment

It is not specified at which timesteps each model is evaluated. Some downstream trajectories contain 21 timesteps, others only 15.

- Inconsistent or unspecified loss functions during fine-tuning

The paper does not state which loss functions (e.g., MSE, relative L1, or hybrid losses) were used for fine-tuning DPOT, POSEIDON, and SPUS. Since the evaluation metric is MSE, using different training losses can lead to misleading cross-model comparisons. If SPUS was trained directly with MSE while the baselines used relative losses (as in their original works), the reported results may unfairly advantage SPUS.

- Absence of AR evaluation for POSEIDON in Appendix results

Figure A.1 in the appendix reports “error growth over time” but only for direct evaluation of POSEIDON. Since AR rollouts are known to substantially reduce long-term error accumulation, this comparison provides little insight into the true relative performance. An additional plot showing AR-based POSEIDON results would be critical for fairness. Moreover, it would be valuable to understand how POSEIDON would perform if it were fine-tuned using the same autoregressive procedure as SPUS. The paper does not clarify whether such a training regime would yield comparable or potentially improved results for POSEIDON.

- No analysis of scaling behavior

A key feature of any foundation model is scaling performance with model size. The paper provides no experiment or discussion on how SPUS behaves when scaled up or down. Without such evidence, it is difficult to assess whether the proposed model exhibits the characteristic scaling trends expected of an FM.

**Questions:**

- How stable is the autoregressive rollout of SPUS for very long horizons (e.g., >50 time steps)? Do the authors observe systematic error accumulation or qualitative drift, and how does it compare to transformer-based models?

- Given that SPUS is trained only to predict discrete next steps, how would the model behave if queried at intermediate time steps ? Would this be even possible in current settings?

- Since SPUS’s design is inherently temporal, how do the authors envision adapting it to stationary or steady-state PDEs where no temporal evolution exists?

- Based on their training experience, do the authors expect SPUS performance to improve with increasing model size and (pretraining) dataset size?

The authors should also review the Weaknesses section for additional (implicit) questions and points raised.

**Details Of Ethics Concerns:**

/

---

> ### Author Response · Authors · 2025-11-21
> **Reply to the Reviewer Part 1**
>
> We appreciate the reviewer’s recognition of the strengths of our work — particularly for acknowledging that convolutional architectures remain competitive for PDE foundation modeling and for highlighting our rigorous evaluation on challenging PDE-Gym benchmarks under realistic autoregressive settings.
>
> ---
>
> ### **Novelty and Conceptual Contribution**
> We thank the reviewer for this constructive comment. Building on the theoretical insights from [1], we have added a detailed analysis in Section 4.2, titled **“Theoretical Foundations of Residual U-Net Efficiency and Generalization in PDE Learning.”** This section provides theoretical reasoning for why the residual U-Net architecture generalizes effectively across diverse PDE families in a more parameter-efficient way than Transformers. We kindly invite the reviewer to refer to this section and share additional feedback on our theoretical treatment of the residual U-Net as a PDE foundation model.
>
> We would like to emphasize that the combination of established components we propose—specifically, incorporating residual connections into a U-Net architecture for PDEs within a foundation model framework—has not been explored in prior work. Most importantly, this proposed design demonstrates strong transferability across diverse PDE families.
>
> In addition, we have added more comprehensive experimental analyses:
> - **Scalability of SPUS** with model size (Appendix A.2 and Table 2)
> - **New comparisons** with unpretrained U-Nets (36M parameters) on downstream datasets (Table 1)
> - **Contribution of individual design components** (Appendix A.3 and Appendix A.4)
> - **Performance analysis of SPUS on time-independent PDEs** (Table 1 and Experiment 5(C))
>
> [1] Model Reduction And Neural Networks For Parametric PDEs
>
> ---
>
> ### **Temporal Generality**
> We thank the reviewer for this valuable comment. To enable continuous predictions from a given time *t* and current state *X*, **SPUS** can be fine-tuned with an adapter that integrates time information into the current state representation, enabling prediction of the output state at that specific time. We have already utilized adapters in our manuscript to address channel discrepancies between downstream PDEs and pretraining PDEs.
>
> ---
>
> ### **Time-Independent PDEs**
> We thank the reviewer for this valuable comment. We have added results for the **SE-AF dataset** in **Table 1** and **Experiment 5(C)**, which is time-independent, demonstrating that **SPUS** can generalize from an autoregressive setting to different time resolutions.
>
> ---
>
> ### **Downstream Evaluation**
> The criteria for selecting downstream tasks are described in **Experiments 5(A)–5(B)**.
>
> - **Experiment 5(A):** Does SPUS generalize to unseen systems governed by the compressible Euler (CE) equations, consistent with its pretraining?
>   → For this criterion, we selected two operators from CE: **CE-RPUI** and **CE-RM**.
>
> - **Experiment 5(B):** Does SPUS generalize to systems governed by equations different from those used in pretraining?
>   → For this criterion, we selected three operators from the Navier–Stokes (NS) equations and one operator from the wave equation: **NS-PwC**, **NS-SL**, **FNS-KF**, and **Wave-Gauss**.
>
> ---
>
> ### **Comparison with POSEIDON**
> In the **POSEIDON** paper (Appendix D.6.2) [2], the authors show that the model’s performance depends on the PDE type and explicitly note:
>
> > “Figure 47 shows for the NS-PwC and the Wave-Layer downstream tasks how the error behaves when using direct or (homogeneous) autoregressive rollouts. We can directly see that it depends very much on the task at hand, as autoregressive rollout works better for the NS-PwC task, whereas direct lead-time input works better for Wave-Layer; this seems to be very dataset- and dynamics-dependent. We therefore choose the best strategy for each task, which is listed in Table 6.”
>
> As reported in **Table 6** of the POSEIDON paper, the authors use **direct prediction for nine datasets** and **autoregressive prediction for six datasets** when comparing results with other models. Moreover, POSEIDON reports the **median relative L1 error only at the final time step** of each downstream task.
>
> In contrast, during **SPUS** evaluation, we consider all time steps—from the first to the last—for downstream tasks. We observed that, under this evaluation setting, POSEIDON performs better with direct prediction. Therefore, we used **direct prediction** for POSEIDON in our downstream comparisons.
>
> [2] Poseidon: Efficient Foundation Models for PDEs

---

> > ### Author Response · Authors · 2025-11-21
> > **Reply to the Reviewer Part 2**
> >
> > ### **Smaller Baseline Variants**
> > We thank the reviewer for this valuable comment. To strengthen our claim of parameter efficiency and architectural simplicity, we have added new comparisons with **unpretrained U-Nets (36M parameters)** on downstream datasets, as shown in **Table 1**. These results demonstrate that **SPUS** achieves higher accuracy compared to unpretrained U-Nets, validating the effectiveness of our pretrained U-Net–based design.
> >
> > ---
> >
> > ### **Evaluation Protocol and Dataset Alignment**
> > We have clearly specified the timesteps at which each model is evaluated. During **SPUS** evaluation, we consider all time steps—from the first to the last—for downstream tasks. This information is provided in **Section 4.4: Finetuning Strategies and Baseline Models** and is also mentioned in the caption of **Table 1**.
> >
> > ---
> >
> > ### **Loss Functions During Fine-Tuning**
> > We thank the reviewer for this valuable comment. All models have been fine-tuned using the **MSE loss**, as now specified in **Section 4.4: Finetuning Strategies and Baseline Models**.
> >
> > ---
> >
> > ### **AR Evaluation for POSEIDON in Appendix**
> > We have added an additional plot showing **AR-based POSEIDON** results in the Appendix.
> > In the **POSEIDON** paper (Appendix D.6.2), the authors show that the model’s performance depends on the PDE type and explicitly note:
> >
> > > “Figure 47 shows for the NS-PwC and the Wave-Layer downstream tasks how the error behaves when using direct or (homogeneous) autoregressive rollouts. We can directly see that it depends very much on the task at hand, as autoregressive rollout works better for the NS-PwC task, whereas direct lead-time input works better for Wave-Layer; this seems to be very dataset- and dynamics-dependent. We therefore choose the best strategy for each task, which is listed in Table 6.”
> >
> > ### **Scaling Behavior**
> > We thank the reviewer for this insightful comment. To address the reviewer’s point on model scalability, we have conducted a detailed analysis of **SPUS** on three downstream datasets using model sizes of **9M**, **36M**, and **76M** parameters, as presented in **Appendix A.2** and **Table 2**. The results show that increasing model size consistently reduces test error across all datasets, demonstrating that **SPUS** scales effectively with model capacity. Notably, the scalability of **U-Net–based architectures** has also been demonstrated in prior work such as [3] **STU-Net: Scalable and Transferable Medical Image Segmentation Models Empowered by Large-Scale Supervised Pre-training.**

---

> > > ### Comment · Reviewer_9Ka3 · 2025-11-26
> > >
> > > To begin, I would like to thank the authors for their additional efforts, revisions, and clarifications.
> > >
> > > - I appreciate the additional analyses in "Novelty and Conceptual Contribution" section. However, my concerns regarding the novelty and the overall contribution of the work remain.
> > >
> > > - My comment on the “loss of temporal generality compared to Poseidon” was intended solely to emphasize that Poseidon possesses additional properties that allow it to be applied both directly and in an autoregressive manner.
> > >
> > > - I thank the authors for adding the time-independent experiment. These results strengthen the paper.
> > >
> > > - For a fair comparison with Poseidon, all downstream tasks evaluated in that study should also be included in your experimental setup. I also believe there is still a discrepancy between the autoregressive inference setup for downstream tasks reported in Poseidon paper and the setup you have used for testing in this work.
> > >
> > > - I appreciate the inclusion of a U-Net baseline. However, a crucial point remains unaddressed: the missing comparison to smaller baseline variants of Poseidon and DPOT. The paper emphasizes that SPUS is parameter-efficient, yet there is no comparison with state-of-the-art foundation models of a similar parameter scale.
> > >
> > > -I thank the authors for adding the scaling behavior study. This is a very valuable contribution for our community.
> > >
> > > While some points and previously unclear details have been clarified, I consider several crucial issues to remain unresolved, in particular the comparison to smaller baseline models, the concerns regarding the novelty, and the omission of a substantial subset of the Poseidon benchmarks. Consequently, I intend to maintain my current score unless these points are adequately addressed.

---

### Official Review · Reviewer_iuzp · 2025-10-28

**Soundness:** 3
**Presentation:** 2
**Contribution:** 2
**Rating:** 4
**Confidence:** 3

**Summary:**

The paper introduces SPUS, a lightweight and parameter-efficient foundation model for solving a broad range of PDE systems. SPUS adopts a residual U-Net architecture with only 36M parameters. It is pretrained autoregressively to mimic numerical solvers and fine-tuned on unseen PDE systems.

**Strengths:**

1. The design is novel and simple, which demonstrates that a lightweight U-Net can serve as an PDE foundation model, challenging the transformer-dominant paradigm.

2. The model achieves competitive results with only one-third the parameters of existing FMs.

3. The model successfully transfers from compressible Euler to incompressible Navier–Stokes and wave equations.

**Weaknesses:**

1. The paper lacks theoretical or mechanistic analysis explaining why a residual U-Net architecture generalizes across diverse PDE families, beyond the observed empirical results.

2. The contribution of individual design components (e.g., residual blocks, autoregressive training, adapters) remains unclear without ablation studies.

3. The experiments are limited to 2D PDEs at fixed resolution. It is not evident whether SPUS can scale to 3D domains or higher spatial resolutions.

4. Including comparisons with smaller non-transformer baselines (e.g., CNNs, FNOs, or unpretrained U-Nets) would strengthen the claim of parameter efficiency and architectural simplicity.

**Questions:**

1. Could the authors provide more theoretical insights into why a residual U-Net architecture can generalize effectively across PDE families with distinct dynamics?

2. Have the authors conducted any ablation analyses to isolate the effect of key design components on the final performance?

3. How well does SPUS scale to 3D PDEs or higher spatial resolutions? Are there architectural or computational bottlenecks that would limit such extensions?

4. Would the authors consider adding smaller CNN-based or operator-based baselines to better substantiate the claim of parameter efficiency?

---

> ### Author Response · Authors · 2025-11-21
>
> We thank the reviewer for recognizing the strengths of our work — particularly our lightweight ResNet-based U-Net design, which serves as a foundation model (FM) for PDEs, challenges the transformer-dominant paradigm, and demonstrates strong transferability across diverse PDEs.
>
>
> ### **Theoretical or Mechanistic Analysis**
> We thank the reviewer for the thoughtful and constructive feedback. Following theoretical insights from [1], we have added a detailed analysis in Section 4.2, titled **“Theoretical Foundations of Residual U-Net Efficiency and Generalization in PDE Learning.”** This new section explains why a residual U-Net architecture can generalize effectively across diverse PDE families. We kindly invite the reviewer to review this section and share further feedback on our theoretical treatment of the residual U-Net as a PDE foundation model.
>
> [1] Model Reduction And Neural Networks For Parametric PDEs
>
>
>
> ### **Contribution of Individual Design Components**
> We appreciate the reviewer’s suggestion to clarify the contribution of individual design components. We have added new analyses in **Appendix A.3**, which quantify the effects of residual blocks on performance of **SPUS** and show that increasing their number improves accuracy in downstream PDE trajectory prediction. In **Appendix A.4**, we further explain how autoregressive training under a first-order Markov process replaces expensive numerical simulators and reduces inference costs for PDE trajectory prediction.
>
>
>
> ### **Parameter Efficiency and Architectural Simplicity**
> We thank the reviewer for this valuable suggestion. To strengthen our claim of parameter efficiency and architectural simplicity, we have added new comparisons with unpretrained U-Nets (36M parameters) on downstream datasets, as shown in **Table 1**. These results demonstrate that **SPUS** achieves higher accuracy compared to unpretrained U-Nets, validating the effectiveness of our pretrained U-Net–based design.

---

> > ### Author Response · Authors · 2025-12-02
> > **Answer for question "How well does SPUS scale to higher spatial resolutions than those used in pretraining?"**
> >
> > We thank the reviewer for this insightful question. To evaluate the scalability of SPUS to spatial resolutions higher than those used during pretraining ($128\times128$), we directly finetuned SPUS-36M on 128 PDE trajectories with a spatial resolution of $256\times256$, governed by the Burgers equation. The dataset was taken from [1]. Each trajectory contains 20 time steps, with the velocity field discretized on a $256\times256$ grid.
> >
> > For comparison, we also trained a 36M-parameter U-Net—identical in architecture to SPUS—from scratch using the same 128 trajectories. The corresponding results have been added to Appendix A.8 (see Figure A.12).
> >
> > Our findings show that SPUS exhibits strong scalability when finetuned on PDE trajectories at a $256\times256$ resolution, despite being pretrained at a lower resolution of $128\times128$.
> >
> > [1] Masked Autoencoders are PDE Learners.

---

### Official Review · Reviewer_XF5w · 2025-10-29

**Soundness:** 2
**Presentation:** 2
**Contribution:** 1
**Rating:** 2
**Confidence:** 4

**Summary:**

This paper presents SPUS, the Small PDE U-Net Solver, which aims to explore the U-Net performance in the PDE foundation model. This paper includes extensive experiments to verify its questions, such as whether SPUS generalize to unseen initial conditions or equations or not, as well as the dataset scalability. In the input-one-frame-rollout-inference setting, SPUS surpassed DPOT and Poseidon.

**Strengths:**

-	It is good to see the investigation of U-Net performance in PDE solving.

-	The authors provide detailed and well-organized experiments.

**Weaknesses:**

Despite the above strengths, this paper contains some unfair experiments that may render the experimental results meaningless.

### (1) Unfair comparison.

As the authors list in section 3, there are three different forecasting settings. Especially, the correct setting in DPOT is based on several past observations to predict the future. I do not think the setting of repeating ts_0 is a correct usage of DPOT.

Also, according to the statement in “even though DPOT was pretrained on operators of both compressible and incompressible NS equations”, I think the authors directly use the pre-trained models provided by DPOT and did not align the pre-training data with SPUS. This is also quite unfair to baselines, since both pre-training and evaluation data are from PDEgym.

### (2) Limited novelty.

Although I acknowledge that the authors attempt to rethink the previous architecture of the PDE foundation model, I cannot appreciate the novelty of this paper since this is just an experiment of pre-training a U-Net with PDE data.

All the analyses are just visualizations or quantitative results. I do not think this paper elaborates on why the U-Net works and why it works in a more parameter-efficient way than Transformers.

### (3) About the scalability experiments.

It is common sense that Transformers usually present log-log scalability, which involves both parameter and dataset aspects and can be rigorously tested based on extensive scaling experiments. However, from Table 2, I cannot justify the scalability of SPUS. I think the authors should follow this paper [1] for further experiments. In my opinion, I do not believe that U-Net has good scalability.

[1] Scaling Laws for Neural Language Models. Tech report OpenAI 2020.

[2] Training Compute-Optimal Large Language Models. Tech report DeepMind 2022.

### (4) Missing relative work.

Actually, DPOT and Poseidon are not state-of-the-art foundation models. Please compare with the following work [3].

[3] Unisolver: PDE-Conditional Transformers Are Universal PDE Solvers, ICML 2025.

### (5) Limitation in irregular geometries.

The current design is limited to regular geometries. I think the authors should discuss this in the limitations section.

**Questions:**

Do the authors adopt the same pre-training data for all the compared baselines?

---

> ### Author Response · Authors · 2025-11-21
>
> We thank the reviewer for recognizing the strengths of our work — particularly the investigation of U-Net performance in PDE solving.
> ### 1. Comparison
> We thank the reviewer for this valuable comment. The goal of **SPUS** is to autoregressively predict the states of PDEs from an initial trajectory state, serving as a numerical solver without relying on expensive neural simulators during inference. Since DPOT requires 10 previous timesteps to predict the next state, we follow the same autoregressive prediction strategy described in the Poseidon paper [1] (see Section D.5 in their work for DPOT results) and compare our results with DPOT accordingly.
>
> We also follow the common practice of utilizing pretrained foundation models and fine-tuning them on downstream datasets, as Poseidon did for DPOT and DPOT did for MPP-L [2] in their respective comparisons. Notably, DPOT is pretrained on ten diverse PDE datasets, including Navier–Stokes (NS), diffusion–reaction, and shallow-water equations. In contrast, **SPUS** is not exposed to the Navier–Stokes dataset during pretraining yet still achieves better results than DPOT on unseen NS downstream tasks, demonstrating the strong transferability and potential of **SPUS** as a PDE foundation model.
>
> [1] Poseidon: Efficient Foundation Models for PDEs
> [2] Multiple Physics Pretraining for Spatiotemporal Surrogate Models
>
> ### 2. Novelty
> We thank the reviewer for the thoughtful and constructive feedback. Building on the theoretical insights from [3], we have added a detailed analysis in Section 4.2, titled **“Theoretical Foundations of Residual U-Net Efficiency and Generalization in PDE Learning.”** This section provides theoretical reasoning for why the **residual U-Net** architecture generalizes effectively across diverse PDE families in a more parameter-efficient way than Transformers. We kindly invite the reviewer to refer to this section and share additional feedback on our theoretical treatment of the **residual U-Net** as a PDE foundation model.
>
> [3] Model Reduction And Neural Networks For Parametric PDEs
>
> ### 3. Scalability
> We thank the reviewer for this insightful comment. To address the point on model scalability, we conducted a detailed analysis of **SPUS** on three downstream datasets using model sizes of 9M, 36M, and 76M parameters, as presented in **Appendix A.2** and **Table 2**. The results show that larger model sizes consistently reduce test error across all datasets, demonstrating that **SPUS** scales effectively with model capacity. Notably, the scalability of **U-Net–based architectures** has also been observed in prior work, such as [4] **STU-Net: Scalable and Transferable Medical Image Segmentation Models Empowered by Large-Scale Supervised Pre-training.**
>
> ### 4. Related Work
> As **Unisolver: PDE-Conditional Transformers Are Universal PDE Solvers (ICML 2025)** is a multimodal model that takes the current PDE state along with equation symbols, coefficients, and boundary conditions, and is a concurrent work, we did not include it in our comparisons.
>
> ### 5. Irregular Geometries
> We thank the reviewer for this valuable observation. We agree that the current design is limited to regular geometries. We have now discussed this limitation in **Section 6 (Limitations).**

---

> > ### Comment · Reviewer_XF5w · 2025-11-27
> >
> > I would like to thank the authors for their feedback and rebuttal. My concerns about novelty and scalability are partially resolved. Thus, I raise my score to 4.
> >
> > However, I still do not think the comparison w.r.t. DPOT is fair, since you modified their prediction paradigm. The idea of following the numerical solver prediction paradigm is reasonable, but the numerical solver has an exact equation, while your model does not contain such information. I think SPUS does not "inference" but only memory the dataset bias. That is why many related works rely on the previous observation for future prediction, which is more convincing.

---

### Meta-Review · Area_Chair_Q6r2 · 2026-01-06

**Summary:**

This manuscript considers foundation models for solving PDEs. The main point is to explore the U-Net architecture for foundation models, which is significantly simpler than existing approaches often based on transformers. While the experiments demonstrate some promise, overall the metareviewer feels that the paper does not contain sufficient novelty to be considered publication.

**Reviewer Concerns:**

Lack of comparison with other foundation models; lack of significant novelty.

**Reviewer Scores:**

While the reviewer XF5w has indicated the possibility of raising the score, the average rating would be still far below the bar of acceptance.

---

### Decision · Program_Chairs · 2026-01-26

Reject